



# Characterization of gas-phase organics using proton transfer
# reaction time-of-flight mass spectrometry: fresh and aged
# residential wood combustion emissions
Emily A. Bruns[a]*, Jay G. Slowik[a], Imad El Haddad[a], Dogushan Kilic[a], Felix Klein[a], Josef
Dommen[a], Brice Temime-Roussel[b], Nicolas Marchand[b], Urs Baltensperger[a] and André S. H.
Prévôt[a]*
*[a]Laboratory of Atmospheric Chemistry, Paul Scherrer Institute, 5232 Villigen, Switzerland*
*[b]Aix Marseille Université, CNRS, LCE UMR 7376, 13331, Marseille, France*
*Correspondence to:* E. A. Bruns (emily.bruns@psi.ch) or A. S. H. Prévôt
(andre.prevot@psi.ch)

13                                      August 21, 2016

15                          For submission to *Atmospheric Chemistry and Physics*



**Abstract**
Organic gases emitted during the flaming phase of residential wood combustion are
characterized individually and by functionality using proton transfer reaction time-of-flight mass
spectrometry.  The evolution of the organic gases is monitored during photochemical aging.
Primary gaseous emissions are dominated by oxygenated species (e.g., acetic acid, acetaldehyde,
phenol and methanol), many of which have deleterious health effects and play an important role
in atmospheric processes such as secondary organic aerosol formation and ozone production.
Residential wood combustion emissions differ considerably from open biomass burning in both
absolute magnitude and relative composition.  Ratios of acetonitrile, a potential biomass burning
marker, to CO are considerably lower (~0.09 pptv ppbv$^{-1}$) than those observed in air masses
influenced by open burning (~1-2 pptv ppbv$^{-1}$), which may make differentiation from
background levels difficult, even in regions heavily impacted by residential wood burning.
Considerable formic acid forms during aging (~200-600 mg kg$^{-1}$ at an OH exposure of (4.5-
5.5)$\times 10^{7}$ molec cm$^{-3}$ h), indicating residential wood combustion can be an important local source
for this acid, the quantities of which are currently underestimated in models.  Phthalic anhydride,
a naphthalene oxidation product, is also formed in considerable quantities with aging (~55-75 mg
kg$^{-1}$ at an OH exposure of (4.5-5.5)$\times 10^{7}$ molec cm$^{-3}$ h).  Although total NMOG emissions vary
by up to a factor of ~9 between burns, SOA formation potential does not scale with total NMOG
emissions and is similar in all experiments.  This study is the first thorough characterization of
both primary and aged organic gases from residential wood combustion and provides a
benchmark for comparison of emissions generated under different burn parameters.



## 1 Introduction


Residential wood combustion is a source of gaseous and particulate emissions in the atmosphere,
including a complex mixture of non-methane organic gases (NMOGs) (McDonald et al., 2000;
Schauer et al., 2001; Hedberg et al., 2002; Jordan and Seen, 2005; Pettersson et al., 2011;
Evtyugina et al., 2014; Reda et al., 2015). NMOGs impact climate (Stocker et al., 2013) and
health (Pouli et al., 2003; Bølling et al., 2009) both directly and through the formation of
products during atmospheric processing (Mason et al., 2001; Kroll and Seinfeld, 2008; Shao et
al., 2009), which makes NMOG characterization critical. Although two studies have speciated a
large fraction of the NMOG mass emitted during residential wood combustion (McDonald et al.,
2000; Schauer et al., 2001), these studies relied on offline chromatographic approaches, which
are time consuming in terms of sample preparation and analysis  and can introduce both positive
and negative artifacts (Nozière et al., 2015). Relatively recently, the proton transfer reaction
mass spectrometer (PTR-MS) has emerged as a powerful tool for online quantification of
atmospherically-relevant NMOGs (Lindinger et al., 1998; Jordan et al., 2009) eliminating many
of the artifacts associated with offline approaches. NMOGs emitted during open burning of a
variety of biomass fuels in the laboratory have been recently quantified using a high resolution
proton transfer reaction time-of-flight mass spectrometer (PTR-ToF-MS) (Stockwell et al., 2015)
and select nominal masses were followed during aging of residential wood combustion emissions
using a quadrupole PTR-MS (Grieshop et al., 2009a). However, a complete high-resolution
characterization of residential wood combustion emissions has yet to be performed.
The quantities and composition of NMOGs emitted during residential wood combustion are
highly dependent on a number of parameters including wood type, appliance type and burn
conditions, and as few studies have characterized these NMOGs (McDonald et al., 2000; Schauer





et al., 2001; Hedberg et al., 2002; Jordan and Seen, 2005; Pettersson et al., 2011; Evtyugina et
al., 2014; Reda et al., 2015), further work is needed to constrain emission factors, as highlighted
in the recent review article by Nozière et al. (2015).  Also, little is known about the evolution of
NMOGs from residential wood combustion with aging.
In this study, we present results from the first use of a smog chamber and a PTR-ToF-MS to
characterize primary and aged gaseous emissions from residential wood combustion in real-time.
This novel approach allows for an improved characterization of NMOG emissions, particularly
oxygenated NMOGs, which are a considerable fraction of the total NMOG mass emitted during
residential wood combustion (McDonald et al., 2000; Schauer et al., 2001).  This study focuses
on a narrow set of burn conditions, namely the flaming phase of beech wood combustion, in
order to generate as reproducible emissions as possible for a complementary investigation of the
effects of parameters such as temperature on the emissions.  While these experiments are a
narrow representation of real-world conditions, this novel work provides a benchmark and
direction for future wood combustion studies.
**2 Methods**
**2.1 Emission generation and smog chamber operation**
Beech (*Fagus sylvatica*) logs are combusted in a residential wood burner (Avant, 2009, Attika)
and emissions are sampled from the chimney through a heated line (473 K), diluted by a factor of
~8-10 using an ejector diluter (473 K, DI-1000, Dekati Ltd.) and injected into the smog chamber
(~7 $m^3$) through a heated line (423 K).  Emissions are sampled during the stable flaming phase of
the burn and modified combustion efficiencies (MCEs), defined as the ratio between $CO_2$ and the
sum of CO and $CO_2$, range from 0.974-0.978 (Table 1).



Emissions are injected for 11-21 min and total dilution factors range from ~100-200. All
experiments are conducted under similar conditions with starting wood masses in the burner of
2.9±0.3 kg and a wood moisture content of 19±2%. The smog chamber has an average
temperature of 287.0±0.1 K and a relative humidity of 55±3% over all five experiments. After
characterization of the primary emissions, as described below, a single dose of d9-butanol (2 µl,
butanol-D9, 98%, Cambridge Isotope Laboratories) is injected into the chamber and a continuous
injection of nitrous acid in air (2.3-2.6 l min$^{-1}$, ≥99.999%, Air Liquide) into the chamber begins.
The decay of d9-butanol measured throughout aging is used to estimate hydroxyl radical (OH)
exposures (Barmet et al., 2012). Nitrous acid produces OH upon irradiation in the chamber and
is used to increase the degree of aging. Levels of $NO_x$ in the chamber prior to aging range from
~150-350 ppb. The small continuous dilution in the chamber during aging due to the constant
nitrous acid injection is accounted for using CO as an inert tracer. The chamber contents are
irradiated with UV light (40 lights, 90-100 W, Cleo Performance, Philips) for 4.5-6 h (maximum
OH exposures of (4.7-6.8)×10$^7$ molec cm$^{-3}$ h which corresponds to ~2-3 days of aging in the
atmosphere at an OH concentration of 1×10$^6$ molec cm$^{-3}$). Reported quantities of aged species
are taken at OH exposures of (4.5-5.5)×10$^7$ (Table 1; ~1.9-2.3 days of aging in the atmosphere at
an OH concentration of 1×10$^6$ molec cm$^{-3}$) (Barmet et al., 2012).
**2.2 Gas-phase analysis**
NMOGs with a proton affinity greater than that of water are measured using a PTR-ToF-MS
(PTR-ToF-MS 8000, Ionicon Analytik GmbH) and $CO_2$, CO and $CH_4$ are measured using cavity
ring-down spectroscopy (G2401, Picarro, Inc.). The PTR-ToF-MS operates with hydronium ion
([$H_2O+H$]$^+$) as the reagent, a drift tube pressure of 2.2 mbar, a drift tube voltage of 543 V and a
drift tube temperature of 90°C leading to a ratio of the electric field (*E*) and the density of the



buffer gas (*N*) in the drift tube (reduced electric field, *E/N*) of 137 Townsend (Td).  The
transmission function is determined using a gas standard of six NMOGs of known concentration
(methanol, acetaldehyde, propan-2-one, toluene, *p*-xylene, 1,3,5-trimethylbenzene; Carbagas).
As the RH and temperature of the sampled air is similar in all experiments, changes in the
detection efficiency of individual species are not expected.
PTR-ToF-MS data are analyzed using the Tofware post-processing software (version 2.4.5,
TOFWERK AG, Thun, Switzerland; PTR module as distributed by Ionicon Analytik GmbH),
running in the Igor Pro 6.3 environment (version 6.3, Wavemetrics Inc.).  The minimum
detection limit is taken as three standard deviations above the background, where the standard
deviation is determined from the measurements of each ion in the chamber prior to emission
injection.  Isotopic contributions are constrained during peak fitting and are accounted for in
reported concentrations.   Possible molecular formulas increase with increasing *m/z*, making
accurate peak assignments difficult in the higher *m/z* range.  Mass spectral data from *m/z* 33 to
*m/z* 130 are assigned molecular formulas, as well as the $^{18}$O isotope of the reagent ion and signal
above *m/z* 130 corresponding to compounds previously identified during residential wood
combustion (McDonald et al., 2000; Schauer et al., 2001; Hedberg et al., 2002; Jordan and Seen,
2005; Pettersson et al., 2011; Evtyugina et al., 2014; Reda et al., 2015).  All signal above *m/z* 130
is included in total NMOG mass quantification.  Using this approach, ~94-97% of the total
NMOG mass measured using the PTR-ToF-MS has an ion assignment.
The reaction rate constant of each species with the reagent ion in the drift tube is needed to
convert raw signal to concentration.  When available, individual reaction rate constants are
applied to ions assigned a structure (Cappellin et al., 2012) (Table S1), otherwise a default
reaction rate constant of $2\times10^{-9}$ cm$^3$ s$^{-1}$ is applied.  For possible isomers, the reaction rate



constant is taken as the average of available values.  Approximately 60-70% of the total NMOG
mass is comprised of compounds with known rate constants.  NMOG signal is normalized to
$[H_2^{18}O+H]^+$ to convert to concentration.  Emission factors (EFs) normalize concentrations to the
total wood mass burned (e.g., mg kg$^{-1}$ reads as mg of species emitted per kg wood burned) to
facilitate comparison between experiments and are calculated as described previously (Andreae
and Merlet, 2001; Bruns et al., 2015a).
PTR-ToF mass spectrometry is a relatively soft ionization technique generally resulting in
protonation of the parent NMOG ($[M+H]^+$), although some compounds are known to produce
other ions, for example through fragmentation or rearrangement (e.g., Baasandorj et al. (2015)).
Reactions potentially leading to considerable formation of species besides $[M+H]^+$ are discussed
in the Supplement.  The extent to which reactions leading to ions other than $[M+H]^+$ occurs is
dependent on instrument parameters such as $E/N$.  The unknown relative contributions of various
isomers makes it difficult to account for reactions generating ions besides $[M+H]^+$ and thus, no
fragmentation corrections are applied.  Emission factors of compounds likely to undergo
extensive reaction to form products besides $[M+H]^+$ (i.e., methylcyclohexane (Midey et al.,
2003), ethyl acetate (Baasandorj et al., 2015) and saturated aliphatic aldehydes (Buhr et al.,
2002), with the exception of acetaldehyde) are not reported.  Due to interferences, butenes
($[C_4H_8+H]^+$) are not quantified.
**3  Results and Discussion**
**3.1 NMOG emissions**
In all experiments, the largest EFs for a single gas-phase species correspond to $CO_2$ (1770-1790
g kg$^{-1}$) and CO (27-30 g kg$^{-1}$) (Table 2), which are in good agreement with previous





measurements from residential beech logwood combustion where $CO_2$ EFs of ~1800 g kg$^{-1}$ and
CO EFs of ~20-70 g kg$^{-1}$ were measured (Ozil et al., 2009; Schmidl et al., 2011; Kistler et al.,
2012; Evtyugina et al., 2014; Reda et al., 2015).  Methane is also emitted in considerable
quantities (1.5-2.8 g kg$^{-1}$), similar to previously observed values for beech wood burning in
fireplaces (0.5-1 g kg$^{-1}$ (Ozil et al., 2009), however, at generally lower levels than total NMOGs
(1.5-13 g kg$^{-1}$).  Total NMOG EFs from beech wood combustion have not been previously
reported, but values are similar to studies of residential wood stove burning of different
hardwoods which have attempted a detailed quantification of total NMOGs, such as McDonald
et al. (2000) (6.2-55.3 g kg$^{-1}$ for a hardwood mixture) and Schauer et al. (2001) (6.7 g kg$^{-1}$ for
oak).  Total NMOG quantities reported in this study refer to species quantified using the PTR-
ToF-MS.
Although a large fraction of atmospherically-relevant organic gases are measured using the PTR-
ToF-MS, some species are not quantitatively detected, including those with a proton affinity less
than water (i.e., small alkanes).  Based on previous studies of residential burning, alkanes are
estimated to contribute less than ~5% to the NMOG mass of either hard or softwood and the sum
of alkenes and alkynes, some of which are quantifiable with the PTR-ToF-MS, are estimated to
contribute less than ~15% to the total measured NMOG mass (McDonald et al., 2000; Schauer et
al., 2001).
Figure 1 shows the primary NMOG mass spectrum for each experiment classified by NMOG
functionality and the fractional contribution of NMOG functional groups to the total NMOG
mass.  EFs for individual compounds are presented in Table 2.  For ease of reading, nominal *m/z*s
are presented in the text and figures, however, monoisotopic *m/z*s for all identified species can be
found in Tables 2 and S2.  Separation of isobaric species is possible using the PTR-ToF-MS,





however, isomers remain indistinguishable.  Quantities of gas-phase species generated during
residential wood combustion depend on a variety of parameters, such as type of burner and wood
species.  However, many compounds are commonly emitted and structures are assigned to
observed ions based on previously identified species (McDonald et al., 2000; Schauer et al.,
2001; Hedberg et al., 2002; Jordan and Seen, 2005; Pettersson et al., 2011; Evtyugina et al.,
2014; Reda et al., 2015).  A few small, unambiguous ions are also assigned a structure, including
methanol, formic acid and acetonitrile.  Approximately 70% of the total NMOG mass measured
using the PTR-ToF-MS is assigned a structure based on this method.
NMOGs are categorized by functional groups including: oxygenated, total $C_xH_y$, nitrogen-
containing and other.  Oxygenated subcategories include: acids (comprised of non-aromatic
acids), carbonyls (comprised of non-aromatic carbonyls), oxygenated aromatics (not including
furans), furans, O-containing (comprised of structurally unassigned oxygenated compounds and
multifunctional oxygenated compounds) and O- and N-containing (comprised of species
containing both oxygen and nitrogen atoms).  Species categorized as N-containing contain no
oxygen atoms.  Total $C_xH_y$ subcategories include: aromatic hydrocarbons, and non-aromatic and
structurally unassigned species (referred to as $C_xH_y$ in the text and figures).  Higher molecular
weight species lacking an ion assignment are categorized as "other".  In the case of possible
isomers, ions are categorized according to the species most likely to dominate based on previous
studies (McDonald et al., 2000; Schauer et al., 2001; Hedberg et al., 2002; Jordan and Seen,
2005; Pettersson et al., 2011; Evtyugina et al., 2014; Reda et al., 2015).
Oxygenated species contribute ~68-94% to the total primary NMOG mass, which has important
atmospheric implications due to the role of these compounds in photochemical reactions, for
example by altering $O_3$ and peroxide formation (Mason et al., 2001; Shao et al., 2009).





McDonald et al. (2000) and Schauer et al. (2001) previously observed the dominance of
oxygenated NMOGs during residential burning of other wood types, whereas Evtyugina et al.
(2014) found that benzene and benzene derivatives contributed 59% to the total measured
NMOGs, compared to only 26% from oxygenated compounds for residential burning of beech
wood in a woodstove.  However, Evtyugina et al. (2014), as well as McDonald et al. (2000) and
Schauer et al. (2001), did not include emissions from all lower molecular weight NMOGs, such
as acetic acid.  Oxygenated NMOGs are also reported as a large fraction of NMOGs emitted
during open burning of many biomass fuels (Gilman et al., 2015; Stockwell et al., 2015).
Acids are the most abundant subclass of species in all experiments with an average EF of
$2000 \pm 2000$ mg kg$^{-1}$ and acetic acid ($[C_2H_4O_2+H]^+$ at nominal $m/z$ 61) is the most highly emitted
compound in all experiments.  In addition to acetic acid, $[C_2H_4O_2+H]^+$ can correspond to
glycolaldehyde, however, Stockwell et al. (2015) found that acetic acid contributes ~75-93% to
$[C_2H_4O_2+H]^+$ during open burning of black spruce (*Picea mariana)* and ponderosa pine (*Pinus*
*ponderosa*) and thus, it is expected that this ion is also largely attributable to acetic acid in the
current study.  Acetic acid and formic acid ($[CH_2O_2+H]^+$ at nominal $m/z$ 47) are the most
abundant carboxylic acids in the atmosphere and are important contributors to atmospheric
acidity (Chebbi and Carlier, 1996).  However, the sources of these acids are poorly understood
(Paulot et al., 2011) and data on their EFs from residential wood combustion are relatively
unknown.  The high acetic acid EFs found here indicate that residential wood combustion can be
an important local source of this acid.  Interestingly, the enhancement of acetic acid ($\Delta C_2H_4O_2$)
over background levels relative to CO enhancement ($\Delta CO$) in the current study ranges from ~6
to 80 pptv ppbv$^{-1}$ (Table 1), which is much higher than the average 0.58 pptv ppbv$^{-1}$ (sum of gas
and aerosol phase) measured in an Alpine valley heavily impacted by residential wood





combustion in winter (Gaeggeler et al., 2008). Further work is needed to investigate the source
of this discrepancy, as limited ambient measurements are available from regions heavily
impacted by residential wood combustion. However, it is possible that the ambient
measurements were dominated by emissions produced during poor burning conditions (e.g.,
starting phase) where CO EFs are expected to be higher than during the stable burning phase
investigated in the current study.
The sum of oxygenated and non-oxygenated aromatic compounds contribute ~7-30% (800±300
mg kg$^{-1}$) to the total primary NMOG mass with benzene ([C$_6$H$_6$+H]$^+$ at nominal $m/z$ 79), phenol
([C$_6$H$_6$O+H]$^+$ at nominal $m/z$ 95), and naphthalene ([C$_{10}$H$_8$+H]$^+$ at nominal $m/z$ 129) as the three
most dominant species. Oxidation products of aromatic species are the largest contributors to
residential wood combustion SOA in this study (Bruns et al., 2016) and both aromatic and
related oxidation products are of interest due to their particularly deleterious effects on health (Fu
et al., 2012).
For the other functional group categories, carbonyl and alcohols contribute ~8-12% (600±600
mg kg$^{-1}$) and ~3-5% (300±300 mg kg$^{-1}$), respectively, to the total NMOG mass. In general, the
most highly emitted carbonyl compound is acetaldehyde ([C$_2$H$_4$O+H]$^+$ at nominal $m/z$ 45).
Methanol ([CH$_3$OH+H]$^+$ at nominal $m/z$ 33) is the most highly emitted alcohol, although other
acyclic alcohols can undergo extensive fragmentation in the mass spectrometer. Furans are only
a minor contributor to the total primary NMOG mass, contributing ~3-5% (300±300 mg kg$^{-1}$),
but are of potential interest as several furans were recently identified as SOA precursors (Gómez
Alvarez et al., 2009) and possible open biomass burning markers (Gilman et al., 2015).
**3.2  Burn variability**



Although the same compounds are emitted during all burns, there is variability in EFs between
experiments despite efforts to replicate burns as closely as possible and the fact that the MCE for
each experiment falls within a narrow range (0.974-0.978) (Table 1). Experiments 2 and 3 show
marked differences in total NMOG EFs and NMOG composition compared to experiments 1, 4
and 5. For example, the total NMOG EF is ~9 times higher in experiment 2 compared to
experiment 5 (Table 2). Acetic acid EFs vary by a factor of ~15 between burns, with high
emissions in experiments 2 and 3 relative to experiments 1, 4 and 5. The total emission of
oxygenated species also correlates with acetic acid emissions, with total oxygenated EFs
considerably higher in experiments 2 and 3 than in experiments 1, 4 and 5. In contrast, aromatic
hydrocarbons and $C_xH_y$ EFs show no correlation with total oxygenated species or acetic acid
EFs. Interestingly, differences in black carbon EFs, primary organic aerosol EFs and primary
organic aerosol mass to black carbon ratios are also not observed between these two groupings of
experiments (2, 3 and 1, 4, 5), as presented previously (Bruns et al., 2016). Enhancements in the
average EF for the different functional groups in experiments 2 and 3 relative to experiments 1, 4
and 5 are shown in Figure 2.
The differences in EFs due to inter-burn variability illustrate the difficulty in constraining EFs
from residential wood combustion. Further work to constrain the possible range of EFs
generated under different conditions is critical for improving model inputs. EFs are also
dependent on factors such as appliance type and fuel loading and further work is needed to
characterize the emissions and the evolution of these emissions with aging generated from
burning of different wood types and under different burn parameters.
**3.3 Biomass burning tracers**



Individual compounds emitted exclusively or in large quantities during biomass burning are of
interest for source apportionment and compounds contributing to SOA formation are of
particular interest for climate and health (Figure 3).  Acetonitrile is used as an ambient gas-phase
marker for open biomass burning (de Gouw et al., 2003; Singh et al., 2003).  In the current
experiments, acetonitrile EFs are relatively low (3.5±0.3 mg kg$^{-1}$) compared to open biomass
burning (~20-1000 mg kg$^{-1}$) (Yokelson et al., 2008; Yokelson et al., 2009; Akagi et al., 2013;
Stockwell et al., 2015), likely due to different burn conditions (e.g., oxygen availability).  The
enhancements of acetonitrile over background levels relative to CO enhancement,
$\Delta CH_3CN/\Delta CO$, are ~0.08-0.1 pptv ppbv$^{-1}$ (Table 1).  This is slightly lower than the only
previously published residential wood combustion measurements (0.1 to 0.8 pptv ppbv$^{-1}$)
(Grieshop et al., 2009a), but is much lower than $\Delta CH_3CN/\Delta CO$ measurements in ambient air
masses impacted by open biomass burning (~1-2 pptv ppbv$^{-1}$) (Holzinger et al., 1999; Andreae
and Merlet, 2001; Christian et al., 2003; de Gouw et al., 2003; Jost et al., 2003; Holzinger et al.,
2005; de Gouw et al., 2006; Warneke et al., 2006; Yokelson et al., 2008; de Gouw et al., 2009;
Yokelson et al., 2009; Aiken et al., 2010; Akagi et al., 2013).  However, $\Delta CH_3CN/\Delta CO$ during
open burning has been shown to depend strongly on fuel type; Stockwell et al. (2015) observed
$\Delta CH_3CN/\Delta CO$ values from 0.0060-7.1 pptv ppbv$^{-1}$ for individual open burns of different
biomass types.  Further work is needed to investigate $CH_3CN$ emissions from residential burning
of other wood types, as well as emissions during other burning phase (e.g., smoldering).
However, these low enhancements may be difficult to differentiate from ambient background
levels, making acetonitrile a poor marker for residential wood combustion.
The interference from isobaric compounds when quantifying acetonitrile using a PTR-MS is an
important consideration when high resolution data are not available.  Previously, several studies





have determined this interference is minimal during open biomass burning (de Gouw et al., 2003;
Warneke et al., 2003; Christian et al., 2004; Warneke et al., 2011).  Recently, Dunne et al. (2012)
quantified interferences with acetonitrile measurements in polluted urban air using a quadrupole
PTR-MS and found contributions of 5-41% to *m/z* 42 from non-acetonitrile ions including:
$[C_3H_6]^+$ and the $^{13}$C isotope contribution from $[C_3H_5]^+$.  In the current study, in addition to
contributions from $[C_3H_6]^+$ and the isotopic contribution from $[C_3H_5]^+$, ~30-50% of the total
signal at *m/z* 42 is due to $[C_2H_2O]^+$, which is presumably a fragment from higher molecular
weight species.  The total contribution to *m/z* 42 from species besides acetonitrile is ~70-85%.
Although an investigation into the effects of the PTR-MS operating conditions (e.g., $[O_2]^+$ signal
from ion source, *E/N* affecting fragmentation) is outside the scope of the current study, the
possibility of considerable non-acetonitrile signal at *m/z* 42 should be taken into consideration
when using nominal mass PTR-MS data to quantify acetonitrile from residential wood
combustion.
Methanol is also used to identify air masses influenced by open biomass burning and
enhancement over background levels relative to CO enhancement ($\Delta CH_3OH/\Delta CO$) is typically
~1-80 pptv ppbv$^{-1}$ in ambient and laboratory measurements of fresh open biomass burning
emissions  (Holzinger et al., 1999; Goode et al., 2000; Andreae and Merlet, 2001; Christian et al.,
2003; Yokelson et al., 2003; Singh et al., 2004; Tabazadeh et al., 2004; Holzinger et al., 2005; de
Gouw et al., 2006; Gaeggeler et al., 2008; Yokelson et al., 2008; Yokelson et al., 2009; Akagi et
al., 2013; Stockwell et al., 2015; Müller et al., 2016).  Here, we find similar values ranging from
~2-20 pptv ppbv$^{-1}$ (Table 1), in agreement with Gaeggeler et al. (2008) who measured a
$\Delta CH_3OH/\Delta CO$ value of 2.16 pptv ppbv$^{-1}$ in an Alpine valley heavily impacted by residential
wood combustion emissions in winter.



### 3.4 Chamber studies of NMOG aging


Previous investigations of aged residential wood combustion emissions have largely focused on
the evolution of the aerosol phase (Grieshop et al., 2009a; Grieshop et al., 2009b; Hennigan et
al., 2010; Heringa et al., 2011; Bruns et al., 2015a; Bruns et al., 2015b; Bruns et al., 2016) and
little is known about the evolution of the gas phase. The evolution of the NMOG functional
group categories with increasing OH exposure is shown in Figure 4. Figure 5 shows the absolute
change in mass spectral signal between the aged and primary NMOG quantities. Although an
increase in NMOG mass could be expected with aging due to oxygenation, total NMOG mass
decreases by ~5-30% at an OH exposure of $(4.6\text{-}5.5)\times10^7$ molec cm$^{-3}$ h relative to the primary
emissions in experiments 1-4, likely due to the conversion of species from the gas to particle
phase, the mass of which increased considerably with aging (Bruns et al., 2016), and the
formation of gas-phase species not quantified here (e.g., formaldehyde). The total NMOG mass
increases slightly, by ~5%, in experiment 5. Quantities of individual NMOGs and NMOG
functional group categories after reaching an OH exposure of $(4.6\text{-}5.5)\times10^7$ molec cm$^{-3}$ h are
presented in Table S2.
Subcategories of oxygenated species behave differently with aging. For example, total quantities
(mg kg$^{-1}$) of oxygenated aromatic species decrease by factors of ~7-15 and furan quantities
decrease by factors of ~4-9, whereas all other oxygenated subcategories, as well as N-containing
species, remain within a factor of 2 of primary values at an OH exposure of $(4.6\text{-}5.5)\times10^7$ molec
cm$^{-3}$ h. Aromatic hydrocarbons and $C_xH_y$ quantities decrease with aging by factors of ~1.5-3.
The large decreases in oxygenated aromatic species and furans illustrate the highly reactive
nature of these species with respect to OH. The evolution of the bulk NMOG elemental
composition during aging is shown in Figure S1 in the Supplement.





In all experiments, formic acid quantities increases considerably with aging (by factors of ~5-
50), as does $[C_4H_2O_3+H]^+$ at nominal $m/z$ 99 (by factors of ~2-3), which likely corresponds to
maleic anhydride, both of which are formed during the oxidation of aromatic species among
other compounds (Bandow et al., 1985; Sato et al., 2007; Praplan et al., 2014). However, the
fragment resulting from the loss of water from maleic acid cannot be distinguished from maleic
anhydride using the PTR-ToF-MS. Formic acid is underestimated in models, likely due to
missing secondary sources (Paulot et al., 2011) and these results indicate that aging of residential
wood combustion emissions can result in considerable secondary formic acid production. The
signal at $m/z$ 149, corresponding to $[C_8H_4O_3+H]^+$, increases by factors of ~2-7 with aging. This
ion likely corresponds to phthalic anhydride, which is a known naphthalene oxidation product
(Chan et al., 2009).
Acetic acid formation has been observed in some ambient, open biomass burning plumes with
aging (Goode et al., 2000; Hobbs et al., 2003; Yokelson et al., 2003), whereas not in others (de
Gouw et al., 2006) and a doubling of $m/z$ 61, likely dominated by acetic acid, was observed
during aging of residential burning emissions in a previous laboratory study (Grieshop et al.,
2009a). In the current study, no increase in the average acetic acid concentration relative to
$CO_{(g)}$ is observed (Table 1). Note that this implies production of secondary acetic acid that
compensates for the expected consumption of ~8-10% of primary acetic acid by reaction with
OH at an OH exposure of $(4.5\text{-}5.5)\times10^7$ molec cm$^{-3}$ h. These results indicate that acetic acid
from residential burning of beech wood is dominated by primary emissions of this species (Table
1). As with acetic acid, there are discrepancies in methanol behavior as open biomass burning
plumes undergo aging (Goode et al., 2000; Yokelson et al., 2003; Tabazadeh et al., 2004;
Holzinger et al., 2005; de Gouw et al., 2006; Akagi et al., 2013). As described by Akagi et al.



(2013), methanol enhancement has been hypothesized to correlate with terpene concentration
and here, methanol remains within ~1-20% of the primary value after exposure to $(4.5\text{-}5.5)\times10^7$
molec cm$^{-3}$ h OH (Table 1), which is expected based on the reaction with OH (Overend and
Paraskevopoulos, 1978) and the low terpene concentrations.
We have previously identified the compounds contributing to the majority of the SOA formed
during these experiments (Bruns et al., 2016).  Figure S2 shows the observed decay of the largest
SOA precursors during aging in the chamber compared to the expected decay based on the OH
concentration in the chamber and the reaction rate with respect to OH.  There is good agreement
between the observed and calculated decay for each compound which supports the structural
assignment of each ion.
As described above, the overall primary emission profiles, as well as total NMOG emissions,
vary considerably for experiments 2 and 3 compared to experiments 1, 4 and 5, with
considerably higher total NMOG emissions in experiments 2 and 3.  To determine the impact of
the high NMOG emission experiments (2 and 3) compared to the lower NMOG emission
experiments (1, 4 and 5) on SOA formation potential, individual SOA precursors with published
SOA yields are investigated (Figure 3).  The SOA formation potential for each of these 18
compounds is determined as the product of the primary EF and the best estimate SOA yield
determined from the literature, as determined previously (Bruns et al., 2016).  The total SOA
formation potential for each experiment is taken as the sum of the individual SOA formation
potentials.  Interestingly, the SOA formation potential is similar in all experiments and the
average enhancement of SOA formation potential in experiments 2 and 3 compared to the
average of experiments 1, 4 and 5 is insignificant (Figure 2), despite the considerably different
total NMOG EFs.



## 4 Conclusions

This study is the first detailed characterization of primary NMOGs from residential wood
combustion using a PTR-ToF-MS and the first investigation of the evolution of the majority of
these NMOGs with aging.  Differences in EFs and profiles between residential burning and open
burning can be considerable and these results illustrate the importance of considering these
emission sources individually.  While total emissions from open burning are much larger than
from residential burning, the societal relevance of residential wood burning emissions is
nontrivial.  A large fraction of open biomass burning derives from wildfires in sparsely
populated regions (Ito and Penner, 2004), whereas residential wood combustion has been shown
to be a major fraction of wintertime submicron organic aerosol in densely populated
communities (Glasius et al., 2006; Krecl et al., 2008; Gonçalves et al., 2012; Guofeng et al.,
2012; Crippa et al., 2013; Herich et al., 2014; Tao et al., 2014; Paraskevopoulou et al., 2015).
Interestingly, MCE does not completely capture inter-burn variability, which is driven by
differences in oxygenated content.  This work clearly shows that measurements of total NMOGs
or total hydrocarbon measurements are insufficient for estimating SOA formation potential from
residential wood combustion.  While this work characterizes the stable burning of beech wood in
a modern woodstove, the composition and quantities of wood combustion emissions are highly
dependent on many factors and further work is needed to characterize the emissions and the
evolution of these emissions with aging generated from burning of different wood types and
under different burn parameters.
**Acknowledgements**





The research leading to these results received funding from the European Community's Seventh
Framework Programme (FP7/2007-2013) under grant agreement no. 290605 (PSI-FELLOW),
from the Competence Center Environment and Sustainability (CCES) (project OPTIWARES)
and from the Swiss National Science Foundation (WOOSHI grant 140590 and starting grant
BSSGI0_155846).  We are grateful to René Richter for technical assistance and to Mike Cubison
for analysis support.

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





**Table 1.** Modified combustion efficiencies, OH exposures of reported aged values (molec cm$^{-3}$ h) and enhancement of select species relative to CO enhancement above background levels (pptv ppbv$^{-1}$)

| parameter | experiment | | | | | average[a] |
|---|---|---|---|---|---|---|
| | 1 | 2 | 3 | 4 | 5 | |
| MCE | 0.975 | 0.978 | 0.977 | 0.974 | 0.978 | 0.976±0.002 |
| OH exposure | $4.5\times10^7$ | $5.5\times10^7$ | $5.3\times10^7$ | $5.2\times10^7$ | $4.7\times10^7$ | - |
| $\Delta CH_3CN_{primary}/\Delta CO$ | 0.079 | 0.11 | 0.099 | 0.077 | 0.082 | 0.09±0.01 |
| $\Delta CH_3CN_{aged}/\Delta CO$ | 0.084 | 0.11 | 0.11 | 0.072 | 0.069 | 0.09±0.02 |
| $\Delta CH_3OH_{primary}/\Delta CO$ | 3.4 | 21 | 11 | 2.4 | 1.5 | 8±8 |
| $\Delta CH_3OH_{aged}/\Delta CO$ | 3.4 | 19 | 11 | 2.5 | 1.8 | 7±7 |
| $\Delta C_2H_4O_{2primary}/\Delta CO$ | 12 | 84 | 57 | 9.8 | 5.9 | 30±30 |
| $\Delta C_2H_4O_{2aged}/\Delta CO$ | 12 | 68 | 48 | 9.4 | 6.5 | 30±30 |

[a]Uncertainties correspond to one sample standard deviation of the replicates.

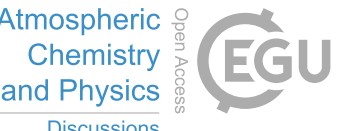

**Table 2.** Primary emission factors of gas-phase species (mg kg$^{-1}$)[a,b]

| species | monoisotopic $m/z$ | structural assignment[c] | functional group | experiment 1 | 2 | 3 | 4 | 5 | average[d] |
|---|---|---|---|---|---|---|---|---|---|
| CO$_2$ | | | | 1780000 | 1781000 | 1777000 | 1772000 | 1784000 | 1779000 ±4000 |
| CO | | | | 27000 | 26000 | 27000 | 30000 | 27000 | 28000±2000 |
| CH$_4$ | | | | 1800 | 1600 | 2000 | 2800 | 1500 | 1900±500 |
| NMOG | | | | 2800 | 13000 | 9200 | 3200 | 1500 | 6000±5000 |
| acid | | | | 750 | 5000 | 3500 | 700 | 340 | 2000±2000 |
| O-containing | | | | 560 | 3400 | 2200 | 590 | 290 | 1000±1000 |
| carbonyl | | | | 310 | 1500 | 960 | 270 | 170 | 600±600 |
| oxygenated aromatic | | | | 230 | 780 | 520 | 270 | 140 | 400±300 |
| alcohol | | | | 130 | 660 | 360 | 90 | 48 | 300±300 |
| furan | | | | 93 | 680 | 410 | 95 | 51 | 300±300 |
| O- and N-containing | | | | 120 | 81 | 77 | 120 | 91 | 100±20 |
| C$_x$H$_y$ | | | | 120 | 210 | 210 | 160 | 64 | 150±60 |
| aromatic hydrocarbon | | | | 320 | 170 | 490 | 680 | 250 | 400±200 |
| N-containing | | | | 20 | 39 | 36 | 23 | 16 | 30±10 |
| other | | | | 140 | 390 | 310 | 160 | 94 | 200±100 |
| [CH$_3$OH+H]$^+$ | 33.034 | methanol | alcohol | 110 | 660 | 360 | 87 | 47 | 300±300 |
| [C$_2$H$_3$N+H]$^+$ | 42.034 | acetonitrile | N-containing | 3.4 | 3.4 | 4.1 | 3.6 | 3.2 | 3.5±0.3 |
| [C$_3$H$_6$+H]$^+$ | 43.055 | propene | C$_x$H$_y$ | 38 | 61 | 40 | 28 | 15 | 40±20 |
| [C$_2$H$_4$O+H]$^+$ | 45.034 | acetaldehyde | carbonyl | 94 | 330 | 230 | 79 | 48 | 200±100 |
| [CH$_2$O$_2$+H]+ | 47.013 | formic acid | acid | 9.9 | 96 | 100 | 31 | 4.2 | 50±50 |
| [C$_2$H$_6$O+H]$^+$ | 47.050 | ethanol | alcohol | 16 | BDL | 3.3 | 2.5 | BDL | 4±7 |
| [C$_4$H$_6$+H]$^+$ | 55.055 | buta-1,3-diene | C$_x$H$_y$ | 14 | 38 | 33 | 14 | 5.7 | 20±10 |
| [C$_3$H$_4$O+H]$^+$ | 57.034 | prop-2-enal | carbonyl | 45 | 160 | 120 | 45 | 25 | 80±60 |
| [C$_2$H$_2$O$_2$+H]$^+$ | 59.013 | oxaldehyde | carbonyl | BDL | BDL | BDL | 1.3 | BDL | 0.3±0.6 |
| [C$_3$H$_6$O+H]$^+$ | 59.050 | propan-2-one propanal | carbonyl | 54 | 190 | 120 | 30 | 30 | 80±70 |
| [C$_2$H$_4$O$_2$+H]$^+$ | 61.029 | acetic acid glycolaldehyde | acid | 740 | 4900 | 3400 | 670 | 340 | 2000±2000 |
| [C$_4$H$_4$O+H]$^+$ | 69.034 | furan | furan | 17 | 140 | 82 | 19 | 9.7 | 50±60 |
| [C$_5$H$_8$+H]$^+$ | 69.070 | isoprene cyclopentene | C$_x$H$_y$ | 3.4 | 12 | 9.4 | 2.8 | 1.1 | 3±2 |
| [C$_4$H$_6$O+H]$^+$ | 71.050 | (E)-but-2-enal 3-buten-2-one 2-methylprop-2-enal | carbonyl | 25 | 120 | 72 | 19 | 14 | 50±40 |
| [C$_5$H$_{10}$+H]$^+$ | 71.086 | (E)-/(Z)-pent-2-ene 2-methylbut-1-ene 2-methylbut-2-ene pent-1-ene 3-methylbut-1-ene | C$_x$H$_y$ | 2.7 | 5.3 | 4.0 | 2.0 | 0.86 | 3±2 |
| [C$_3$H$_4$O$_2$+H]$^+$ | 73.029 | 2-oxopropanal | carbonyl | 26 | 140 | 96 | 26 | 15 | 60±50 |
| [C$_4$H$_8$O+H]$^+$ | 73.065 | butan-2-one butanal 2-methylpropanal | carbonyl | 7.2 | 44 | 24 | 5.2 | 4.2 | 20±20 |
| [C$_3$H$_6$O$_2$+H]$^+$ | 75.045 | methyl acetate | O-containing | 62 | 490 | 300 | 56 | 28 | 200±200 |
| [C$_6$H$_6$+H]$^+$ | 79.055 | benzene | aromatic hydrocarbon | 210 | 90 | 300 | 450 | 150 | 200±100 |
| [C$_5$H$_6$O+H]$^+$ | 83.050 | 2-methylfuran | furan | 21 | 160 | 88 | 21 | 12 | 60±60 |
| [C$_5$H$_8$O+H]$^+$ | 85.065 | 3-methyl-3-buten-2-one | carbonyl | 10 | 69 | 39 | 8.7 | 5.4 | 30±30 |
| [C$_6$H$_{12}$+H]$^+$ | 85.102 | (E)-hex-2-ene 2-methyl-pent-2-ene | C$_x$H$_y$ | BDL | 2.2 | 1.6 | 0.60 | BDL | 1±1 |
| [C$_4$H$_6$O$_2$+H]$^+$ | 87.045 | butane-2,3-dione | carbonyl | 51 | 450 | 250 | 52 | 26 | 200±200 |
| [C$_7$H$_8$+H]$^+$ | 93.070 | toluene | aromatic hydrocarbon | 23 | 22 | 34 | 39 | 16 | 27±9 |
| [C$_6$H$_6$O+H]$^+$ | 95.050 | phenol | oxygenated aromatic | 110 | 110 | 130 | 130 | 68 | 110±20 |
| [C$_5$H$_4$O$_2$+H]$^+$ | 97.029 | furan-2-carbaldehyde | furan | 40 | 270 | 180 | 40 | 21 | 100±100 |
| [C$_6$H$_8$O+H]$^+$ | 97.065 | 2,4-/2,5-dimethylfuran | furan | 11 | 86 | 48 | 11 | 5.5 | 30±30 |
| [C$_4$H$_2$O$_3$+H]$^+$ | 99.008 | maleic anhydride[e] | O-containing | 40 | 91 | 66 | 40 | 26 | 50±50 |
| [C$_8$H$_8$+H]$^+$ | 105.070 | styrene | aromatic hydrocarbon | 12 | 8.0 | 20 | 24 | 9.6 | 15±7 |
| [C$_7$H$_6$O+H]$^+$ | 107.050 | benzaldehyde | oxygenated aromatic | 18 | 14 | 23 | 27 | 11 | 18±7 |
| [C$_8$H$_{10}$+H]$^+$ | 107.086 | m-/o-/p-xylene ethylbenzene | aromatic hydrocarbon | 4.2 | 6.9 | 7.5 | 6.3 | 2.9 | 6±2 |
| [C$_7$H$_8$O+H]$^+$ | 109.065 | m-/o-/p-cresol | oxygenated aromatic | 24 | 71 | 48 | 25 | 14 | 40±20 |
| [C$_6$H$_6$O$_2$+H]$^+$ | 111.045 | m-/o-/p-benzenediol 2-methylfuraldehyde | oxygenated aromatic | 26 | 150 | 86 | 22 | 14 | 60±50 |
| [C$_9$H$_8$+H]$^+$ | 117.070 | 1H-indene | aromatic hydrocarbon | 5.0 | BDL | 9.5 | 15 | 2.9 | 6±6 |
| [C$_9$H$_{10}$+H]$^+$ | 119.086 | 2,3-dihydro-1H-indene | aromatic | 2.3 | 2.8 | 3.9 | 3.3 | 1.3 | 3±1 |



| Ion | m/z | Structural assignment | Category | | | | | | |
|---|---|---|---|---|---|---|---|---|---|
| [C$_8$H$_8$O+H]$^+$ | 121.065 | 1-phenylethanone | oxygenated aromatic | 8.3 | 14 | 13 | 8.8 | 4.6 | 10±4 |
| [C$_9$H$_{12}$+H]$^+$ | 121.102 | 3-/4-methylbenzaldehyde *i*-propylbenzene *n*-propylbenzene | aromatic hydrocarbon | 1.0 | 2.4 | 2.3 | 1.2 | 0.68 | 1.5±0.8 |
| [C$_8$H$_{10}$O+H]$^+$ | 123.081 | 1,3,5-trimethylbenzene 2,4-/2,6-/3,5-dimethylphenol | oxygenated aromatic | 4.7 | 36 | 18 | 4.9 | 3.0 | 10±10 |
| [C$_7$H$_8$O$_2$+H]$^+$ | 125.060 | 2-methoxyphenol methylbenzenediols | oxygenated aromatic | 9.2 | 110 | 55 | 12 | 4.9 | 40±50 |
| [C$_6$H$_6$O$_3$+H]$^+$ | 127.040 | 5-(hydroxymethyl)furan-2-carbaldehyde | furan | 4.4 | 29 | 17 | 4.9 | 2.7 | 10±10 |
| [C$_{10}$H$_8$+H]$^+$ | 129.070 | naphthalene | aromatic hydrocarbon | 42 | 20 | 80 | 100 | 33 | 60±30 |
| [C$_8$H$_{10}$O$_2$+H]$^+$ | 139.076 | 2-methoxy-4-methylphenol 4-(2-hydroxyethyl)phenol | oxygenated aromatic | 3.2 | 59 | 29 | 6.2 | 1.8 | 20±20 |
| [C$_{11}$H$_{10}$+H]$^+$ | 143.086 | 1-/2-methylnaphthalene | aromatic hydrocarbon | 4.0 | 2.3 | 5.7 | 7.5 | 3.3 | 5±2 |
| [C$_9$H$_6$O$_2$+H]$^+$ | 147.045 | 2,3-dihydroinden-1-one | oxygenated aromatic | 11 | 13 | 13 | 11 | 6.0 | 11±3 |
| [C$_8$H$_4$O$_3$+H]$^+$ | 149.024 | phthalic anhdyride$^e$ | O-containing | 16 | 31 | 25 | 16 | 8.3 | 19±9 |
| [C$_8$H$_8$O$_3$+H]$^+$ | 153.055 | 4-hydroxy-3-methoxybenzaldehyde | oxygenated aromatic | 3.8 | 27 | 15 | 3.7 | 1.4 | 10±10 |
| [C$_{12}$H$_8$+H]$^+$ | 153.070 | acenaphthylene | aromatic hydrocarbon | 6.1 | 3.6 | 12 | 15 | 8.3 | 9±5 |
| [C$_9$H$_{12}$O$_2$+H]$^+$ | 153.092 | 4-ethyl-2-methoxyphenol 1,2-dimethoxy-4-methylbenzene | oxygenated aromatic | 1.4 | 30 | 14 | 3.2 | BDL | 10±10 |
| [C$_8$H$_{10}$O$_3$+H]$^+$ | 155.071 | 2,6-dimethoxyphenol | oxygenated aromatic | 2.2 | 73 | 35 | 7.8 | 1.0 | 20±30 |
| [C$_{12}$H$_{10}$+H]$^+$ | 155.086 | 1,1'-biphenyl 1,2-dihydroacenaphthylene | aromatic hydrocarbon | 3.1 | BDL | 4.3 | 6.1 | 2.9 | 3±2 |
| [C$_{12}$H$_{12}$+H]$^+$ | 157.102 | dimethylnaphthalene | aromatic hydrocarbon | 1.3 | 3.0 | 3.2 | 2.2 | 1.2 | 2.2±0.9 |
| [C$_{10}$H$_{12}$O$_2$+H]$^+$ | 165.092 | 2-methoxy-4-[(*E*)-prop-1-enyl]phenol 2-methoxy-4-prop-2-enylphenol 2-methoxy-4-[(*Z*)-prop-1-enyl]phenol | oxygenated aromatic | 0.92 | 24 | 13 | 2.3 | 0.59 | 8±10 |
| [C$_9$H$_{10}$O$_3$+H]$^+$ | 167.071 | 1-(4-hydroxy-3-methoxyphenyl)ethanone 2,5-dimethylbenzaldehyde 3,4-dimethoxybenzaldehyde | oxygenated aromatic | 2.5 | 11 | 6.7 | 2.2 | 1.2 | 5±4 |
| [C$_{13}$H$_{10}$+H]$^+$ | 167.086 | fluorene | aromatic hydrocarbon | BDL | BDL | 1.0 | 2.5 | 2.0 | 1±1 |
| [C$_{10}$H$_{14}$O$_2$+H]$^+$) | 167.107 | 2-methoxy-4-propylphenol | oxygenated aromatic | 0.88 | 7.6 | 4.4 | 1.1 | BDL | 3±3 |
| [C$_9$H$_{12}$O$_3$+H]$^+$ | 169.086 | 2,6-dimethoxy-4-methylphenol | oxygenated aromatic | BDL | 14 | 6.2 | 1.1 | BDL | 4±6 |
| [C$_{14}$H$_{10}$+H]$^+$ | 179.086 | phenanthrene anthracene | aromatic hydrocarbon | 6.4 | 8.4 | 6.1 | 3.6 | 7.7 | 6±2 |
| [C$_{13}$H$_8$O+H]$^+$ | 181.065 | fluoren-9-one phenalen-1-one | oxygenated aromatic | 2.7 | 4.0 | 2.7 | 1.2 | 1.9 | 2±1 |
| [C$_{10}$H$_{12}$O$_3$+H]$^+$ | 181.086 | 1-(4-hydroxy-3-methoxyphenyl)propan-2-one | oxygenated aromatic | BDL | 4.2 | 2.6 | 1.1 | 0.69 | 2±2 |
| [C$_9$H$_{10}$O$_4$+H]$^+$ | 183.066 | 3,4-dimethoxybenzoic acid 4-hydroxy-3,5-dimethoxybenzaldehyde | oxygenated aromatic | 1.1 | BDL | 1.4 | 1.1 | 1.0 | 0.9±0.5 |
| [C$_{10}$H$_{14}$O$_3$+H]$^+$ | 183.102 | 4-ethyl-2,6-dimethoxyphenol | oxygenated aromatic | 1.0 | 7.4 | 4.2 | 1.0 | BDL | 3±3 |
| [C$_{15}$H$_{12}$+H]$^+$ | 193.102 | 1-/2-/3-/9-methylphenanthrene 2-methylanthracene | aromatic hydrocarbon | 0.50 | 2.6 | 1.3 | BDL | 0.44 | 1±1 |
| [C$_{11}$H$_{14}$O$_3$+H]$^+$ | 195.102 | 1,3-dimethoxy-2-prop-2-enoxybenzene 2,6-dimethoxy-4-[(*Z*)-prop-1-enyl]phenol | oxygenated aromatic | BDL | 1.7 | 1.2 | BDL | BDL | 0.6±0.8 |
| [C$_{16}$H$_{10}$+H]$^+$ | 203.086 | fluoranthene pyrene acephenanthrylene | aromatic hydrocarbon | BDL | 0.87 | BDL | BDL | BDL | 0.2±0.4 |

$^a$CO$_2$, CO and CH$_4$ are measured using cavity ring down spectroscopy and all other species are measured using the PTR-ToF-MS.

$^b$BDL indicates value is below the detection limit.

$^c$Multiple structural assignments for a given ion correspond to possible isomers.

$^d$Uncertainties correspond to one sample standard deviation of the replicates.

$^e$Structural assignment based on known products produced during oxidation of aromatics (Bandow et al., 1985; Chan et al., 2009; Praplan et al., 2014).



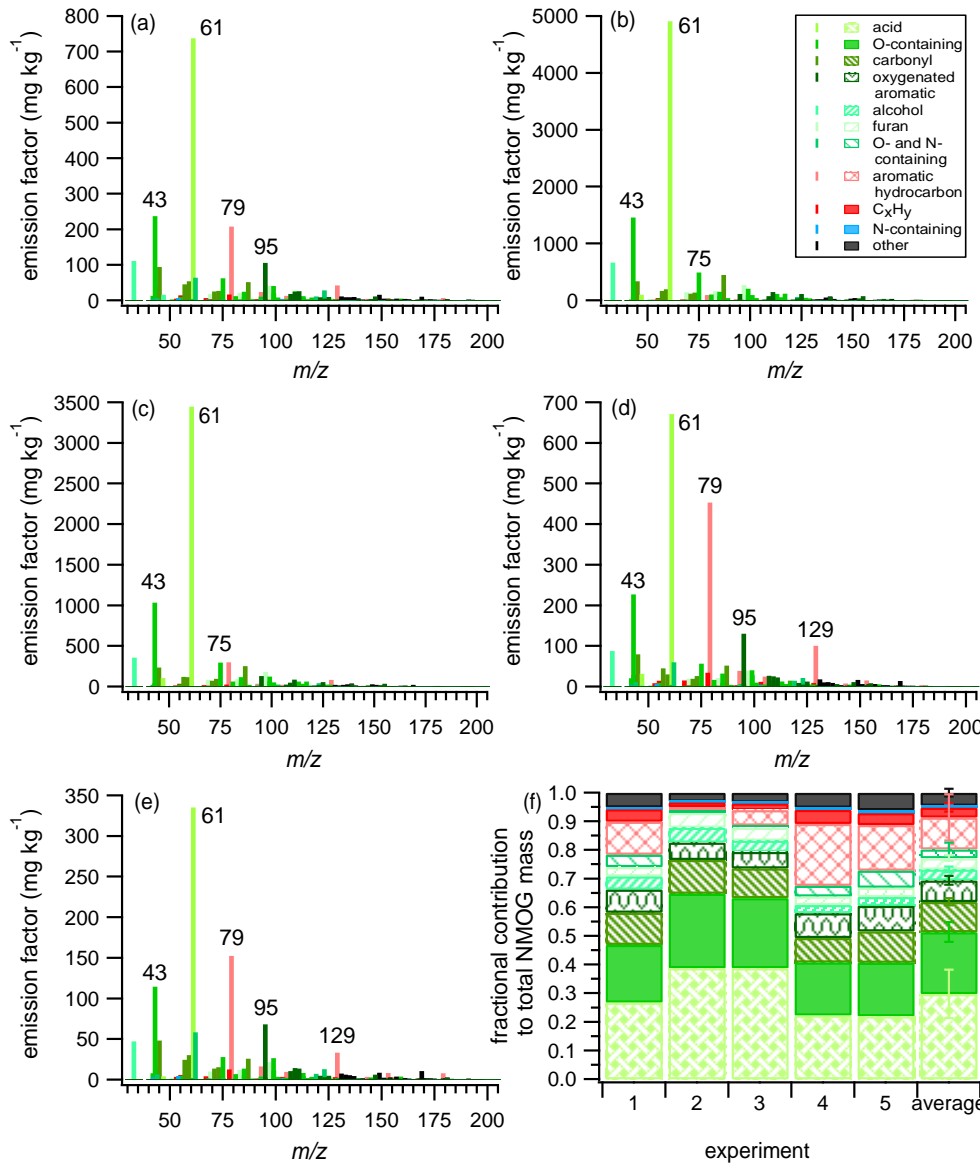

**Figure 1.** Mass spectra of primary emissions for experiments 1-5 (a-e) colored by functional group. (a-e) Labelled peaks correspond to $[C_2H_3O]^+$ (*m/z* 43, fragment from higher molecular weight compounds), $[C_2H_4O_2+H]^+$ (*m/z* 61, acetic acid), $[C_3H_6O_2+H]^+$ (*m/z* 75, methyl acetate), $[C_6H_6+H]^+$ (*m/z* 79, benzene), $[C_6H_6O+H]^+$ (*m/z* 95, phenol) and $[C_{10}H_8+H]^+$ (*m/z* 129, naphthalene). The bars in (f) correspond to the fractional contribution of each functional group to the total NMOG mass for each experiment and the average of all experiments. Error bars correspond to one sample standard deviation of the replicates. Legend in (b) applies to (a-f).





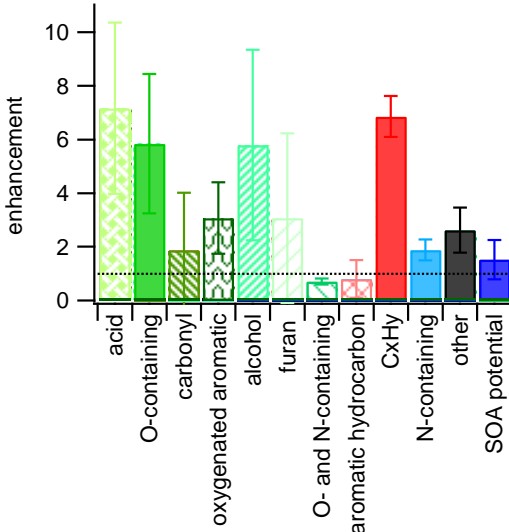

**Figure 2.** Enhancement (average value (mg kg$^{-1}$) of experiments 2 and 3 relative to the average value of experiments 1, 4 and 5) in each NMOG functional group category and for SOA formation potential. Total SOA formation potential is determined using the primary EF of each NMOG identified as a SOA precursor and literature SOA yields and assumes complete consumption of each NMOG with aging (see text for details). Error bars correspond to one sample standard deviation.





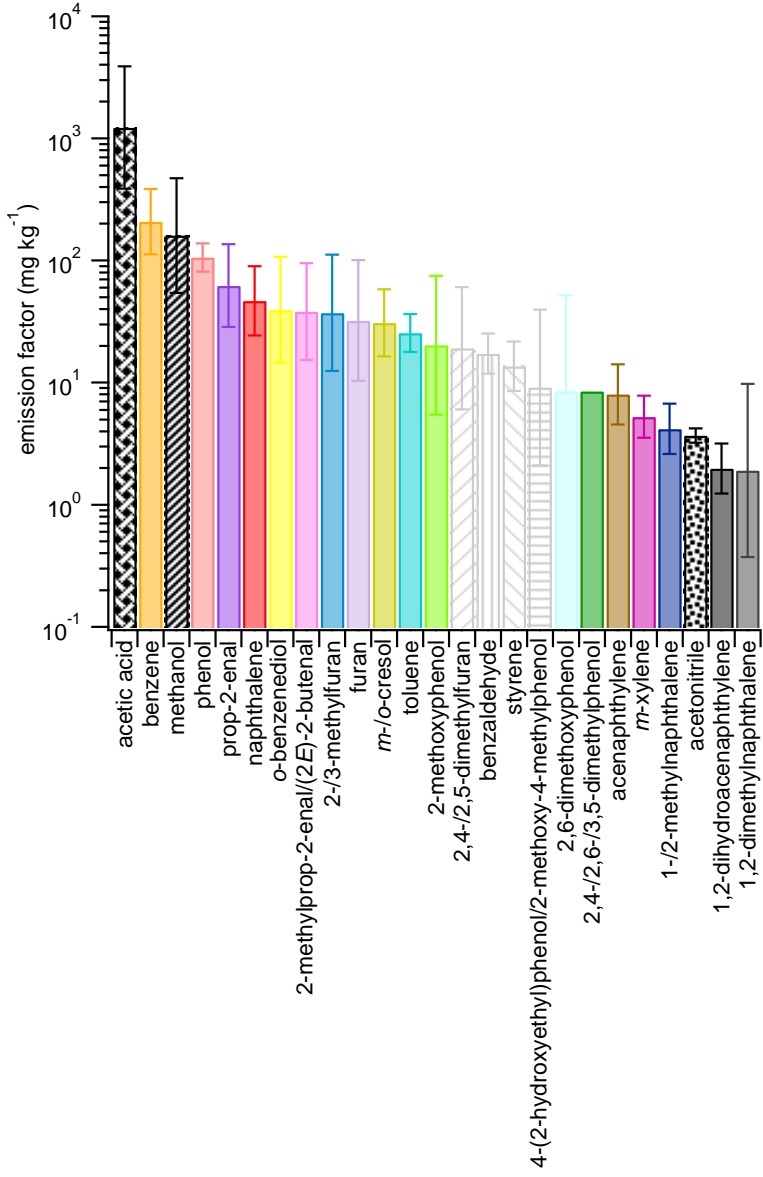

**Figure 3.** Geometric mean of the primary emission factors for gas-phase species of particular interest for SOA formation (solid bars and gray patterned bars) and identification of air masses influenced by biomass burning (black patterned bars). Colors and patterns corresponding to NMOGs contributing to SOA formation are consistent with Bruns et al. (2016). Error bars correspond to the sample geometric standard deviation of the replicates.





**Figure 4.** (a-e) Temporal evolution of gas-phase species categorized by functional group throughout aging in the smog chamber. Units on the y-axes are mass of each functional group (mg) per mass of wood consumed (kg).

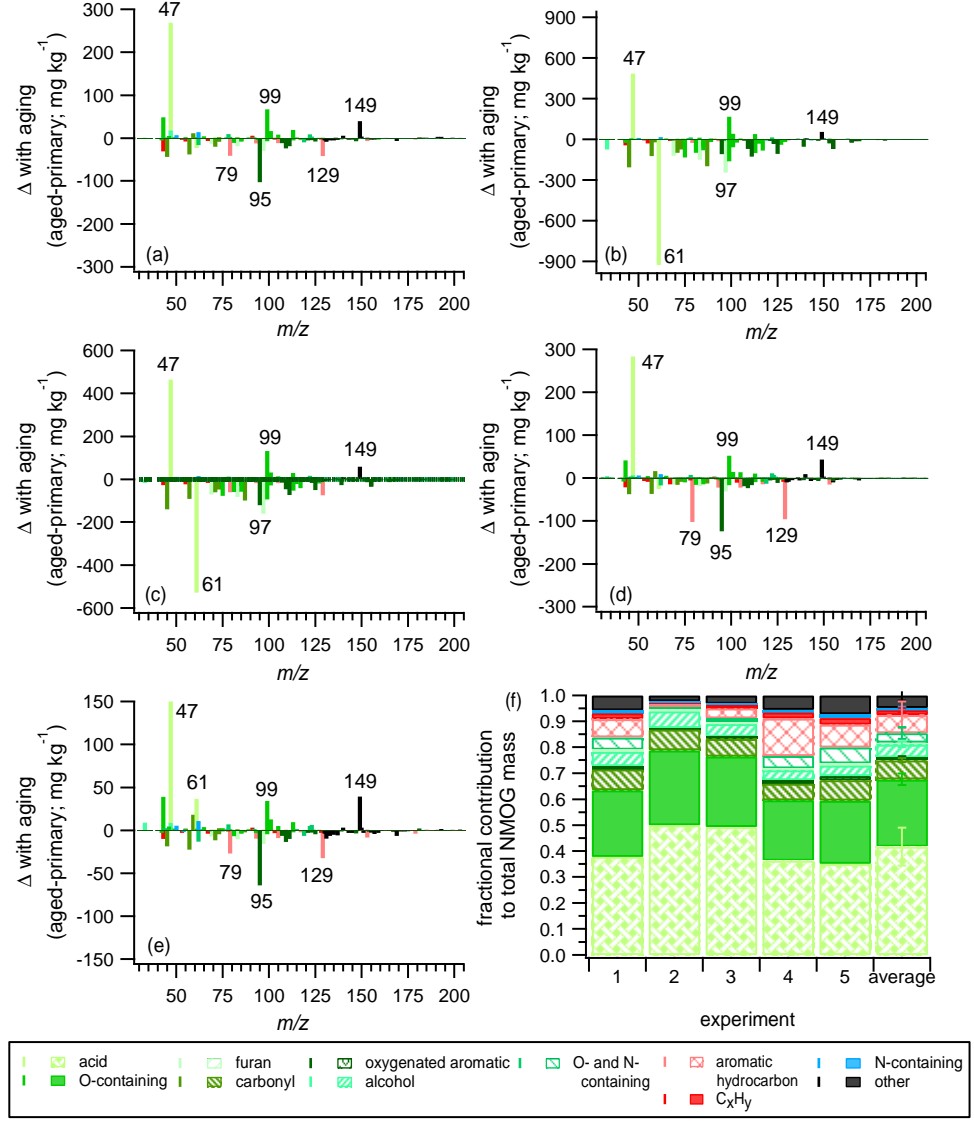

**Figure 5.** Absolute difference of aged and primary mass spectra for experiments 1-5 (a-e), where peaks less than zero decrease during aging and peaks greater than zero increase during aging. Aged emissions correspond to an OH exposure of $(4.5\text{-}5.5)\times10^7$ molec cm$^{-3}$ h. (a-e) Labelled peaks correspond to $[CH_2O_2+H]^+$ (*m/z* 47, formic acid), $[C_2H_4O_2+H]^+$ (*m/z* 61, acetic acid), $[C_6H_6+H]^+$ (*m/z* 79, benzene), $[C_6H_6O+H]^+$ (*m/z* 95, phenol), $[C_5H_4O_2+H]^+$ (*m/z* 97, furan-2-carbaldehyde), $[C_4H_2O_3+H]^+$ (*m/z* 99, maleic anhydride), $[C_{10}H_8+H]^+$ (*m/z* 129, naphthalene) and $[C_8H_4O_3+H]^+$ (*m/z* 149, phthalic anhydride). The bars in (f) correspond to the fractional contribution of each category to the total NMOG EF at an OH exposure of $(4.5\text{-}5.5)\times10^7$ molec cm$^{-3}$ h for each experiment and the average of all experiments. Error bars correspond to one sample standard deviation of the replicates.