# Peer review of "Characterization of gas-phase organics using proton transfer"

_Atmospheric Chemistry and Physics, 2016_

## Referee Comment (RC1) · Anonymous Referee #1 · 9 Sep 2016

Bruns et al. describe controlled laboratory measurements of fresh and aged emissions from the residential combustion of beech wood. The authors generated these emissions using a commercial wood burner. Using a high-resolution proton transfer reaction time-of-flight mass spectrometer, the authors measured primary VOC emissions under stable flaming conditions. For aging experiments, the emissions were directed into a Teflon chamber and oxidized by OH radicals generated from the photolysis of nitrous acid. Primary emissions exhibited significant enhancements of oxygenated species (particularly acids) and aromatic compounds. The emissions of typical nitrogen-containing biomass burning markers, such as acetonitrile, were significantly

lower than those observed from open burning. During aging experiments, the authors observed significant consumption of NMOG mass. Certain species, such as formic acid and phthalic anhydride, showed significant enhancements. Acetic acid, however, exhibited no net increase, which the authors attribute to the balancing of secondary production + OH consumption.

The manuscript is written clearly and the contents are well organized. The study is interesting, well executed, and the results provide insights into the chemical evolution of wood smoke, which is poorly constrained yet important for regional air quality. My primary comments pertain to the conclusions drawn about secondary NMOG and the observations of low acetonitrile. In particular, I believe the authors should provide an expanded discussion (and potentially further insights) into the variability of NMOG oxidation products (see point 2). Upon addressing these comments, I recommend the manuscript for publication.

Comments

1) Secondary NMOG:

The authors discuss a number of processes that could affect the observed net decrease in NMOG mass, including gas-to-particle partitioning and conversion of gas-phase species to those that cannot be detected by the PTR-ToF-MS. However, the authors do not include a discussion about vapor-phase wall loss. Bian et al. (2015) simulated the loss of primary biomass burning emissions to a Teflon chamber and demonstrated that wall loss can significantly affect both particle and gas-phase organics. In the average simulation, $\sim$ 75% of gas-phase vapors were lost to the chamber. Stockwell et al. (2014) observed losses of biomass burning organic compounds (including acetic acid) to surfaces at very different rates. Can the authors estimate and/or discuss the impact of wall loss and potentially provide uncertainties to the $5-30\%$ loss in NMOG mass?

In addition to wall loss, I think the authors should also discuss the variability of secondary organic production. This discussion is provided for primary emissions (Section 3.2), but few insights are drawn from the variability of oxidation products. There are significant differences between the trends observed during Expts. 2,3 and those observed during Expts. 1,4,5 (Figs. 4 and 5). For example, acids and O-containing compounds show a general increase in Expts 1,4,5, but a decrease in Expts 2,3. It is notable that the initial NMOG distributions in Expts 1,4,5 contain a higher fraction of aromatic and oxygenated aromatics. Could it be that these compounds are a significant source of secondary acids and O-containing compounds? It should also be noted that other compounds not measured by proton-transfer could also impact these trends (e.g. ethylene). This variability is quite interesting and a discussion pertaining to these differences may help in understanding the variability of OVOC formation in open burning (e.g. de Gouw et al. 2006 vs Yokelson et al. 2003).

2) Acetonitrile

In Section 3.3, the authors discuss the variability of acetonitrile. The authors attribute the observations of low acetonitrile to burning conditions. While burning efficiency and O2 fraction certainly affect NMOG emissions, very recent work demonstrates that fuel composition plays a major role in the variability of nitrogen-containing VOCs (Coggon et al. 2016). In that study, the authors show that wood (low nitrogen content) emits a significantly lower fraction of nitrogen-containing VOCs than other tree components, such as leaves and boughs (high nitrogen content).

Given this new work, the authors should also discuss the effects of fuel composition. Assuming that the beech wood is free of stems, twigs, or leaves, then it is likely that low acetonitrile emissions result from the combustion of low nitrogen-containing fuel. Have the authors also considered looking at the emissions of other nitrogen-containing NMOGs that are sensitive to proton-transfer, such as acrylonitrile or HNCO? These species would also likely exhibit lower EFs compared to open burning of fuels with higher nitrogen content.

Other Comments

Line 45: The descriptor "residential wood combustion" is unclear. Other studies have investigated the emissions from fuels typically burned in stoves (e.g. Douglas Fir, Stockwell 2015). To avoid confusion, please specify that you are speciating wood combustion emissions from commercial stoves.

Line 76: Please provide more details about the burner. Is the appliance fitted with a catalyst or secondary combustion zone? A description or schematic would be helpful for other researchers studying the emissions from other wood burners.

Line 90-91 What kind of lights are used to photolyze HONO? Can the authors provide flux measurements (or cite a source containing this information)?

Line 91: How do these levels of NOx compare to those from other biomass burning sources? NOx will also depend on fuel composition (e.g. Burling et al. 2010). Furthermore, how do NOx levels change after initiating the photolysis of HONO? Did the authors also measure ozone? If so, how much was formed as a result of photochemical processing? I believe these conditions are important to discuss, especially for future studies focused on biomass burning aging.

Section 3.2. The discussion about burn variability is much appreciated. Can the authors propose reasons for these differences? The tight reproducibility of MCE makes me think it's not necessarily burning efficiency. Could there also be variability in how the burner operates that could lead to these differences (e.g. temperature)? Syc et al. observed significantly different emission factors of PAHs from a commercial burner when burning lignite at various temperatures. Hansson et al. (2004) observed differences in nitrogen NMOG distributions as a function of temperature for the pyrolysis of bark and other biomass sources. I would imagine that similar effects could be true for the combustion of beech wood.

Fig. 4: I assume that each panel is the temporal evolution of gas-phase species from

each aging experiment. Is that correct? Please clarify.

References:

Bian, Q., May, A. A., Kreidenweis, S. M., and Pierce, J. R.: Investigation of particle and vapor wall-loss effects on controlled wood-smoke smog-chamber experiments, Atmos. Chem. Phys., 15, 11027-11045, doi:10.5194/acp-15-11027-2015, 2015.

Coggon, M.M., Veres, P.R., Yuan, B., Koss, A., Warneke, C., Gilman, J.B., Lerner, B., Peischl, J., Aikin, K., Stockwell, C., Hatch, L., Ryerson, T.B., Roberts, J.M., Yokelson, R., and de Gouw, J.: Emissions of nitrogen-containing organic compounds from the burning of herbaceous and arboraceous biomass: fuel composition dependence and the variability of commonly used nitrile tracers, Geophys. Res. Lett., 43, doi: 10.1002/2016GL070562, 2016.

Hansson, K., Samuelsson, J., Tullin, C., and Amand, L.: Formation of HNCO, HCN, and NH3 from the pyrolysis of bark and nitrogen-containing model compounds, Combustion and Flame, 137 (3), 265-277, doi: 10.1016/j.combustflame.2004.01.005, 2004.

Šyc, M., Horák, J., Hopan, F., Krpec, K., Tomšej, T., Ocelka, T., and Pekárek.V, Effect of Fuels and Domestic Heating Appliance Types on Emission Factors of Selected Organic Pollutants, Environ. Sci. Technol., 45 (21), 9427-9434, doi: 10.1021/es2017945, 2011.

---

## Referee Comment (RC2) · Anonymous Referee #2 · 6 Oct 2016

This paper presents measurements of gas-phase species emitted from combustion of beech in a 'modern' residential heating stove in both fresh and aged states. Five replicate laboratory experiments were conducted in which emissions from a steady flaming combustion were sampled into a laboratory smog chamber, from which there were sampled for both a primary characterization period and during oxidation by OH formed via HONO photolysis. The use of replicate experiments is useful, as it assesses the extent to which emissions vary even within narrowly controlled circumstances – as it turns out, quite a bit. Interestingly, many of the emissions of concern (CO, OA, BC) are quite consistent across tests, as is the SOA produced during aging. In contrast,

the composition and emission factors of NMOG varied substantially, with two tests having markedly different NMOG emissions. The evolution of NMOG is described in which many identified species show the expected decay with OH oxidation, while other species (acids, other O-containing species) showed enhancement.

This paper is clearly written and the measurements and analysis appear to be of high quality. This topic is of great interest to the readership of ACP as it provides important insights into the composition and evolution of an important class of biomass burning emissions. Below I have highlighted several points that would like to see addressed in revisions. The main focus of my comments is on the difference between the two sets of experiments – I would like to see a bit more discussion of the conditions that lead to these differences and how the two 'anomalous' experiments (#2, #3) differ from the others. This seems to be a key point, and while the differences are discussed, there's little investigation of what might have influenced this difference. For example, the fuel consumed was nearly half during these experiments than the others, why? One general comment is that I was really required to read the other Bruns et al. 2016 paper in order to understand and interpret these results. While I understand that the authors split these aspects of the reporting of the project to avoid a cumbersome manuscript, I would like to see this one 'stand alone'. So, at the least I would suggest that a table of basic test parameters (like Table 1 in Bruns et al 2016) be included, perhaps in the SI. Another general comment is that there is a bit of an over-emphasis on differences between results observed here (one fuel, one combustion condition) and observations more generally. In most cases, these comparisons are appropriately caveated, but in some cases the generalizations are a bit sweeping (e.g. Line 283) – I ask the authors to give this a once over to ensure that these results, while certainly providing key insights and data, are not over-extrapolated. Finally, I second many of the concerns/questions of the first referee.

Once these general and specific points have been addressed, I recommend the manuscript for publication in ACP.

[Figure]

Specific points

I am a bit confused by Table 2 – this indicates acetonitrile as the only N-containing species, but EF of acetonitrile is ∼10 times lower than that for N-containing species? Is the rest of this mass contributed by un-identified compounds? It also doesn't seem as if N-containing species contribute 20-30 mg/kg to NMOG mass on Figure 1, but it would be hard to see there.

N-containing species were higher for Expts. 2-3, but aerosol-phase nitrate was substantially lower (even accounting for lower fuel consumption) (Bruns et al. 2016). Were there any other notable differences? E.g. NOx levels? Have you examined secondary nitrate formation during aging?

L255 – Figure 2 is just mentioned here, and this could bear a bit more discussion and the differences between experiments discussed a bit further. One thing that stands out about these two experiments (2, 3) is that the CO2 loading in the chamber was substantially (almost half) lower than the other three experiments (despite the fact that the MCEs and many other quantities are essentially the same. If the injection time and dilution conditions were the same, this suggests that perhaps the combustion rate was lower (which would probably be indicated by lower flue temperature). Are there any other contextual or operational details that were different?

L283 – This is too broad/definitive of a statement to make based on the narrow set of conditions tested here.

L317-319 – It would be useful/instructive to attempt a mass balance on the NMOG and SOA loadings to estimate how much of the measured NMOG may be ending up in the condensed phase in your experiments.

L358 – Were terpenes actually quantified? I don't expect much from birch wood, but if you measured them (or found them BDL) this should be noted.

L362 – 'Good agreement' is a bit vague, there is not-great agreement in panels c) and

f). Could this indicate possibly misattribution of these compounds? For this figure, it would be helpful to show smoothed data (and probably a log y-axis) to make this a bit more readable.

L365-370 - Somewhat confusing lead-in to discussion of Figure 3, as discussion emphasizes differences between experiments and this figure shows averages across all experiments - may make sense to just discuss this result then discuss inter-experiment differences (using Fig. 2)

References

Bruns, E. A., El Haddad, I., Slowik, J. G., Kilic, D., Klein, F., Baltensperger, U., and Prévôt, A. S. H. (2016). "Identification of significant precursor gases of secondary organic aerosols from residential wood combustion." Scientific Reports, 6, 27881.

---

## Author Comment (AC1) · 1 Dec 2016

**Response to Referees**

Manuscript acp-2016-753

Characterization of gas-phase organics using proton transfer reaction time-of-flight mass spectrometry: fresh and aged residential wood combustion emissions

Emily A. Bruns, Jay G. Slowik, Imad El Haddad, Dogushan Kilic, Felix Klein, Josef Dommen, Brice Temime-Roussel, Nicolas Marchand, Urs Baltensperger and André S. H. Prévôt

We thank the Referees for the comments and we have incorporated the feedback to improve the manuscript. We have copied the remarks of each Referee in *black italics* and our responses are given in regular black font. Manuscript text including revisions is given in regular blue font.

**Anonymous Referee #1**

Bruns et al. describe controlled laboratory measurements of fresh and aged emissions from the residential combustion of beech wood. The authors generated these emissions using a commercial wood burner. Using a high-resolution proton transfer reaction time-of-flight mass spectrometer, the authors measured primary VOC emissions under stable flaming conditions. For aging experiments, the emissions were directed into a Teflon chamber and oxidized by OH radicals generated from the photolysis of nitrous acid. Primary emissions exhibited significant enhancements of oxygenated species (particularly acids) and aromatic compounds. The emissions of typical nitrogen-containing biomass burning markers, such as acetonitrile, were significantly lower than those observed from open burning. During aging experiments, the authors observed significant enhancements. Acetic acid, however, exhibited no net increase, which the authors attribute to the balancing of secondary production + OH consumption.

The manuscript is written clearly and the contents are well organized. The study is interesting, well executed, and the results provide insights into the chemical evolution of wood smoke, which is poorly constrained yet important for regional air quality. My primary comments pertain to the conclusions drawn about secondary NMOG and the observations of low acetonitrile. In particular, I believe the authors should provide an expanded discussion (and potentially further insights) into the variability of NMOG oxidation products (see point 2). Upon addressing these comments, I recommend the manuscript for publication.

We have addressed the Referee's comments as detailed below, including modifying the manuscript to provide an expanded discussion and further insights into the variability of NMOG oxidation products and observations of low acetonitrile emissions.

**Comments**

**1) Secondary NMOG:**

The authors discuss a number of processes that could affect the observed net decrease in NMOG mass, including gas-to-particle partitioning and conversion of gasphase species to those that

cannot be detected by the PTR-ToF-MS. However, the authors do not include a discussion about vapor-phase wall loss. Bian et al. (2015) simulated the loss of primary biomass burning emissions to a Teflon chamber and demonstrated that wall loss can significantly affect both particle and gas-phase organics. In the average simulation, ~75% of gas-phase vapors were lost to the chamber. Stockwell et al. (2014) observed losses of biomass burning organic compounds (including acetic acid) to surfaces at very different rates. Can the authors estimate and/or discuss the impact of wall loss and potentially provide uncertainties to the 5 - 30% loss in NMOG mass?

NMOG wall losses were inferred by monitoring NMOG concentrations prior to initiating photochemistry and by assessing the smog chamber conditions affecting loss rates during aging as detailed by Zhang et al. (2014 and 2015) (Zhang, X., Cappa, C.D., Jathar, S.H., McVay, R.C., Ensberg, J.J., Kleeman, M.J. and Seinfeld, J.H.: Influence of vapor wall loss in laboratory chambers on yields of secondary organic aerosol, Proceedings of the National Academy of Sciences 111, 5802–5807, 2014; Zhang, X., Schwantes, R. H., McVay, R. C., Lignell, H., Coggon, M. M., Flagan, R. C., and Seinfeld, J. H.: Vapor wall deposition in Teflon chambers, Atmos. Chem. Phys., 15, 4197–4214, 2015). Bian et al. (2015) found that the concentration of gas-phase emissions generated during open burning decreased by 86% due to vapor wall losses in a dark chamber using best estimate parameters in a model. When using effective wall saturation concentrations based on the study of Zhang et al., Bian et al. (2015) found that the net vapor loss to the walls decreased by 65% compared to the best estimate. While open biomass burning emission profiles share similarities with residential burning emissions, differences can be large (e.g., see response below regarding emission of nitrogen-containing species), making it difficult to apply the findings of Bian et al. (2015) to residential burning, particularly as vapor losses are very sensitive to the model parameters. We think more investigation of residential wood combustion emissions is needed to be able to apply meaningful uncertainties on the loss in NMOG mass in the current study, but we agree with the Referee that a discussion of potential NMOG wall losses is needed in the manuscript and we have modified the text as follows (Pages 16-17, lines 349-359): "In addition to gas to particle phase partitioning and formation of gasphase species not quantified here, a decrease in NMOG mass with aging could also be due to losses of gas-phase species to the chamber walls (Zhang et al., 2014; Bian et al., 2015). Measurements of NMOGs in the chamber prior to aging are stable, indicating that the chamber walls are not a sink for NMOGs, but rather that NMOGs are in equilibrium with the chamber walls, particles and the gas phase. Zhang et al. (2014) show that the rate of NMOG wall loss is proportional to seed aerosol concentration and OH concentration, both of which were relatively high in the current experiments (Table S1; OH concentrations were  $\sim 1.4 \times 10^7$  molec cm-3). Under these experimental conditions, NMOG wall losses are not expected to be large. However, future studies are needed to provide insight into vapor wall loss of residential wood combustion emissions during aging."

In addition to wall loss, I think the authors should also discuss the variability of secondary organic production. This discussion is provided for primary emissions (Section 3.2), but few insights are drawn from the variability of oxidation products. There are significant differences between the trends observed during Expts. 2,3 and those observed during Expts. 1,4,5 (Figs. 4 and 5). For example, acids and O-containing compounds show a general increase in Expts 1,4,5, but a decrease in Expts 2,3. It is notable that the initial NMOG distributions in Expts 1,4,5

contain a higher fraction of aromatic and oxygenated aromatics. Could it be that these compounds are a significant source of secondary acids and O-containing compounds? It should also be noted that other compounds not measured by proton-transfer could also impact these trends (e.g. ethylene). This variability is quite interesting and a discussion pertaining to these differences may help in understanding the variability of OVOC formation in open burning (e.g. de Gouw et al. 2006 vs Yokelson et al. 2003).

We agree with the Referee that more discussion on the variability of the aged emission profiles is needed and we have added a new section in the manuscript to address this topic (3.5 Aged emission variability). We have also included the discussion on variability of SOA formation potential in this section. The text has been modified as follows (Pages 19-20, section 3.5): "As described above, the primary emission profiles, as well as total NMOG mass emitted, vary considerably for experiments 2 and 3 compared to experiments 1, 4 and 5, with much higher total NMOG emissions in experiments 2 and 3. It is expected that the aged emission profiles also exhibit variability based on the primary emissions. Total acid and O-containing species decrease with aging in experiments 2 and 3, in contrast to experiments 1, 4 and 5, where these classes increase with aging (Figure 4). Formic acid shows the largest increase with aging in all experiments (~190-480 mg kg-1 relative to the primary EF, Figure 5), however, in experiments 1, 4 and 5, this increase contributes much more to the total acid mass as the total acid mass is ~5-15 times lower compared to experiments 2 and 3. An analogous case occurs for maleic anhydride for the O-containing class of compounds. As formic acid and maleic anhydride are formed from the oxidation of aromatic compounds (Bandow et al., 1985; Sato et al., 2007; Praplan et al., 2014), among others, a higher fraction of aromatic species to the total NMOG emissions will contribute to increases in acid and O-containing NMOGs. Inclusion of NMOGs not quantified by PTR-ToF-MS could impact the trends observed in Figure 4."

**2) Acetonitrile**

In Section 3.3, the authors discuss the variability of acetonitrile. The authors attribute the observations of low acetonitrile to burning conditions. While burning efficiency and O2 fraction certainly affect NMOG emissions, very recent work demonstrates that fuel composition plays a major role in the variability of nitrogen-containing VOCs (Coggon et al. 2016). In that study, the authors show that wood (low nitrogen content) emits a significantly lower fraction of nitrogen-containing VOCs than other tree components, such as leaves and boughs (high nitrogen content).

Given this new work, the authors should also discuss the effects of fuel composition. Assuming that the beech wood is free of stems, twigs, or leaves, then it is likely that low acetonitrile emissions result from the combustion of low nitrogen-containing fuel. Have the authors also considered looking at the emissions of other nitrogen-containing NMOGs that are sensitive to proton-transfer, such as acrylonitrile or HNCO? These species would also likely exhibit lower EFs compared to open burning of fuels with higher nitrogen content.

We thank the Referee for bringing the recent work of Coggon et al. (2016) to our attention. The beech wood in our study was free of stems, twigs, leaves and bark, and based on the work of Coggon et al. (2016), we therefore expect that a relatively low fraction of the total NMOGs was N-containing compared to burning of biomass containing leaves, etc. This fact may explain the relatively low acetonitrile emissions in our study compared to open biomass burning, where leaves, bark, etc. are typically present. The primary emission factors of  $C_3H_3N$  and HNCO

ranged in our study from 3.6-6.4 mg kg-1 and BLD (<tens of pptv)-11 mg kg-1, respectively. Emission factors of acrylonitrile (C3H3N) observed during open burning are higher than those observed in the current study (~10-90 mg kg-1, Akagi et al., 2013), as expected based on the lower acetonitrile emission factors observed in the current study and the findings of Coggon et al., 2016.

We have expanded the discussion of acetonitrile emissions to include this information (Page 14, lines 288-300): "In agreement with the current study, ambient measurements of acetonitrile made in Colorado (USA) were not associated with fresh residential burning emissions (Coggon et al., 2016). Lower ambient measurements of nitrogen-containing NMOGs (including acetonitrile) during residential burning compared to open burning were attributed to the generally lower nitrogen content in fuels burned residentially (Coggon et al., 2016). Lower nitrogen content of the fuel is likely a contributor to the relatively low acetonitrile emissions in the current study.

The primary emission factors of other nitrogenated species, such as  $C_3H_3N$  (likely corresponding to acrylonitrile) and HNCO ranged in our study from 3.6-6.4 mg kg-1 and BDL-11 mg kg-1, respectively. Emission factors of  $C_3H_3N$  in the current study are lower than those observed during open burning (e.g., ~10-90 mg kg-1 (Akagi et al., 2013)), as expected based on the lower acetonitrile emission factors observed in the current study and the findings of Coggon et al. (2016)."

**Other Comments**

Line 45: The descriptor "residential wood combustion" is unclear. Other studies have investigated the emissions from fuels typically burned in stoves (e.g. Douglas Fir, Stockwell 2015). To avoid confusion, please specify that you are speciating wood combustion emissions from commercial stoves.

The text has been modified as follows to specify that wood combustion emissions were speciated from commercial stoves (Page 3, lines 44-46): "Although two studies have speciated a large fraction of the NMOG mass emitted during residential wood combustion in commercial burners..."

Line 76: Please provide more details about the burner. Is the appliance fitted with a catalyst or secondary combustion zone? A description or schematic would be helpful for other researchers studying the emissions from other wood burners.

The burner was manufactured in 2009 and is not fitted with a catalyst or other emission control device. There is no secondary combustion zone. A photograph of the burner was added to the SI (new Figure S1) and a description of the burner was added to the main text (Page 4, lines 77-78): "... a residential wood burner (Figure S1; single combustion chamber, operated in single batch mode; Avant, 2009, Attika)..."

Line 90-91 What kind of lights are used to photolyze HONO? Can the authors provide flux measurements (or cite a source containing this information)?

In the chamber, HONO is photolyzed using 40 UV lights of 90-100 W (Cleo Performance, Philips) (Page 5, lines 98-99). Emission spectra of these lights, as well as inferred NO2 amd HONO photolysis rates for a similar set-up to the current study, can be found in Platt et al. (2013): Secondary organic aerosol formation from gasoline vehicle emissions in a new mobile environmental reaction chamber, Atmos. Chem. Phys. 13, 9141-9158, 2013. This reference has been added to the manuscript.

Line 91: How do these levels of NOx compare to those from other biomass burning sources? NOx will also depend on fuel composition (e.g. Burling et al. 2010). Furthermore, how do NOx levels change after initiating the photolysis of HONO? Did the authors also measure ozone? If so, how much was formed as a result of photochemical processing? I believe these conditions are important to discuss, especially for future studies focused on biomass burning aging.

For NOx, the primary EFs ranged from ~0.5-0.7 g kg-1 (~160-350 ppbv in the chamber (mainly NO); no primary measurement available for experiment 1), which are much lower than literature for open burning (Stockwell et al., 2015 from open biomass burning of ponderosa pine (~2-5 g kg-1) and black spruce (4-5 g kg-1)). Lower NOx (and N-containing NMOGs) EFs are expected due to the lower nitrogen content of the fuel used compared to open burning (Coggon et al., 2016). Upon aging, NOx increased to ~250-380 ppbv after reaching OH exposures of ~(4.5-5.5)×107 molec cm-3 h, due to HONO photolysis. We have to note that NOx was measured using a chemiluminescence analyzer, and therefore the aforementioned concentrations should be considered as upper estimates, as the measurements are affected by NOy species (especially nitric acid).

Considering these high NOx values and the levels of measured reactive NMOGs in the beginning of the experiments (NMOG/NOx ratios of ~1-10), O3 production is favored. For these experiments, we did not measure O3 concentrations. However, previous measurements conducted under similar conditions indicate an initial O3 production with aging. After an initial increase, O3 concentrations significantly decrease due to the decrease of NMOG/NOx ratios (NMOG consumption and NOx increase with HONO photolysis).

The primary NOx values were added to the manuscript in a new table in the SI (Table S1), which includes other experimental parameters, as suggested by the other Referee. The following was added to the main text (Page 5, line 88): "Experimental parameters and primary emission values are summarized in Table S1." and (Page 5, lines 94-96): "Levels of NOx in the chamber prior to aging range from ~160-350 ppbv and increases to ~250-380 ppbv after reaching OH exposures of ~(4.5-5.5)×107 molec cm-3 h (NOx data unavailable for experiment 1)."

Section 3.2. The discussion about burn variability is much appreciated. Can the authors propose reasons for these differences? The tight reproducibility of MCE makes me think it's not necessarily burning efficiency. Could there also be variability in how the burner operates that could lead to these differences (e.g. temperature)? Syc et al. observed significantly different emission factors of PAHs from a commercial burner when burning lignite at various temperatures. Hansson et al. (2004) observed differences in nitrogen NMOG distributions as a function of temperature for the pyrolysis of bark and other biomass sources. I would imagine that similar effects could be true for the combustion of beech wood.

We agree with the Referee that MCE, which is very similar in all experiments, is unlikely to be the cause of the difference in emission profiles between the experiments. As great care was taken to replicate each burn as closely as possible (e.g., similar starting wood mass, number of logs/kindling pieces and wood arrangement prior to ignition) and experimental conditions (e.g., dilution factors), there is no obvious explanation for the inter-experimental variability. The burner was housed in an uninsulated building and, as suggested, the variability could be due to effects of differences in outdoor temperature on the burner which would influence the combustion rate. Each fire was allowed to burn for 15-20 minutes prior to injecting emissions into the smog chamber, which allowed the burner to warm up, however, we did not make temperature measurements in the burner or chimney and temperature differences may have remained. The discussion on inter-burn variability was expanded to include these points (Page 13, lines 262-266): "The burner is housed in an uninsulated building and the emission profile variability could be due to effects of outdoor temperature variability on the burner. For example, emission profiles from burning lignite and pyrolysis of bark and other biomass sources have been shown to vary with burn temperature (Hansson et al., 2004; Šyc et al., 2011)."

**Fig. 4: I assume that each panel is the temporal evolution of gas-phase species from each aging experiment. Is that correct? Please clarify.**

Each panel corresponds to the temporal evolution for a single experiment. The figure legend has been modified, "...Temporal evolution of gas-phase species categorized by functional group throughout aging in the smog chamber for experiments 1-5 (a-e)."

**References:**

Bian, Q., May, A. A., Kreidenweis, S. M., and Pierce, J. R.: Investigation of particle and vapor wall-loss effects on controlled wood-smoke smog-chamber experiments, Atmos. Chem. Phys., 15, 11027-11045, doi:10.5194/acp-15-11027-2015, 2015.

Coggon, M.M., Veres, P.R., Yuan, B., Koss, A., Warneke, C., Gilman, J.B., Lerner, B., Peischl, J., Aikin, K., Stockwell, C., Hatch, L., Ryerson, T.B., Roberts, J.M., Yokelson, R., and de Gouw, J.: Emissions of nitrogen-containing organic compounds from the burning of herbaceous and arboraceous biomass: fuel composition dependence and the variability of commonly used nitrile tracers, Geophys. Res. Lett., 43, doi: 10.1002/2016GL070562, 2016.

Hansson, K., Samuelsson, J., Tullin, C., and Amand, L.: Formation of HNCO, HCN, and NH3 from the pyrolysis of bark and nitrogen-containing model compounds, Combustion and Flame, 137 (3), 265-277, doi: 10.1016/j.combustflame.2004.01.005, 2004.

Šyc, M., Horák, J., Hopan, F., Krpec, K., Tomšej, T., Ocelka, T., and Pekárek.V, Effect of Fuels and Domestic Heating Appliance Types on Emission Factors of Selected Organic Pollutants, Environ. Sci. Technol., 45 (21), 9427-9434, doi: 10.1021/es2017945, 2011.

**Anonymous Referee #2**

This paper presents measurements of gas-phase species emitted from combustion of beech in a 'modern' residential heating stove in both fresh and aged states. Five replicate laboratory experiments were conducted in which emissions from a steady flaming combustion were sampled into a laboratory smog chamber, from which there were sampled for both a primary characterization period and during oxidation by OH formed via HONO photolysis. The use of replicate experiments is useful, as it assesses the extent to which emissions vary even within narrowly controlled circumstances – as it turns out, quite a bit. Interestingly, many of the emissions of concern (CO, OA, BC) are quite consistent across tests, as is the SOA produced during aging. In contrast, the composition and emission factors of NMOG varied substantially, with two tests having markedly different NMOG emissions. The evolution of NMOG is described in which many identified species show the expected decay with OH oxidation, while other species (acids, other O-containing species) showed enhancement.

This paper is clearly written and the measurements and analysis appear to be of high quality. This topic is of great interest to the readership of ACP as it provides important insights into the composition and evolution of an important class of biomass burning emissions. Below I have highlighted several points that would like to see addressed in revisions. The main focus of my comments is on the difference between the two sets of experiments – I would like to see a bit more discussion of the conditions that lead to these differences and how the two 'anomalous' experiments (#2, #3) differ from the others. This seems to be a key point, and while the differences are discussed, there's little investigation of what might have influenced this difference. For example, the fuel consumed was nearly half during these experiments than the others, why? One general comment is that I was really required to read the other Bruns et al. 2016 paper in order to understand and interpret these results. While I understand that the authors split these aspects of the reporting of the project to avoid a cumbersome manuscript, I would like to see this one 'stand alone'. So, at the least I would suggest that a table of basic test parameters (like Table 1 in Bruns et al 2016) be included, perhaps in the SI. Another general comment is that there is a bit of an over-emphasis on differences between results observed here (one fuel, one combustion condition) and observations more generally. In most cases, these comparisons are appropriately caveated, but in some cases the generalizations are a bit sweeping (e.g. Line 283) – I ask the authors to give this a once over to ensure that these results, while certainly providing key insights and data, are not over-extrapolated. Finally, I second many of the concerns/questions of the first referee.

**Once these general and specific points have been addressed, I recommend the manuscript for publication in ACP.**

We have expanded the discussion on the differences between the two sets of experiments (2 and 3 vs 1, 4 and 5) as described in detail below. We agree that this manuscript should 'stand-alone' and have taken the suggestion of the Referee to add a table with the experimental parameters to the SI (new Table S1). We have also modified the text to ensure the insights from these data are not over-extrapolated, as described below. The responses to the concerns/questions of the first Referee are detailed above.

**Specific points**

I am a bit confused by Table 2 – this indicates acetonitrile as the only N-containing species, but EF of acetonitrile is ~10 times lower than that for N-containing species? Is the rest of this mass contributed by un-identified compounds? It also doesn't seem as if N-containing species contribute 20-30 mg/kg to NMOG mass on Figure 1, but it would be hard to see there. N-containing species were higher for Expts. 2-3, but aerosol-phase nitrate was substantially lower (even accounting for lower fuel consumption) (Bruns et al. 2016). Were there any other notable differences? E.g. NOx levels? Have you examined secondary nitrate formation during aging?

There are 14 N-containing species which contribute to this category and it is correct that 13 of these N-containing species have not been structurally assigned (the exception is acetonitrile). One reason for the lack of assignments is a scarcity of published data on N-containing emissions from residential wood combustion compared to emissions of other classes of compounds. However, an educated guess can be made about several of these compounds based on reasonable structures (e.g., C3H3N likely corresponds to acrylonitrile, as discussed in the response to the other Referee). A brief discussion and the range of emission factors observed for C3H3N and HNCO (in the O- and N-containing category), two compounds of interest in open biomass burning emissions, have been added to the text (Page 14, lines 288-300; see response to other Referee). Future work to identify more N-containing species emitted during residential wood combustion would be informative, similar to the recent work on identifying N-containing emissions from open biomass burning (i.e., Coggon et al., 2016, Stockwell et al., 2015). The majority of the mass contributing to the N-containing class is distributed among several compounds and the total mass of N-containing species is the lowest of all classes, which is likely why it is difficult to see these individual N-containing species in Figure 1.

N-containing species were higher and aerosol phase nitrate lower in experiments 2 and 3 compared to experiments 1, 4 and 5, however, there was no notable difference in NOx emissions between experiments 2 and 3 (~0.5 and 0.7 g kg-1, respectively) compared to experiments 1, 4 and 5 (~0.5 and 0.6 g kg-1, respectively; NOx data not available for experiment 1). With aging, particulate nitrate  $(NO_3)$  showed varied behavior; however, there is no trend between experiments 2 and 3 compared to experiments 1, 4 and 5. After correction for wall losses, NO3 remains stable in experiment 1, NO3 increases by  $\sim 15\%$  of its primary value during initial aging and then remains stable in experiment 2, NO3 decreases by  $\sim 10\%$  of its primary value during initial aging and then remains stable in experiment 3, and NO3 increases by  $\sim 15\%$  of its primary value in experiments 4 and 5 and then slowly decreases with further aging. AMS measurements of NO3 includes inorganic and organic species; however, characterization, including quantification, of organic nitrate species is challenging using aerosol mass spectrometry and more work is needed to investigate particulate organic nitrates from residential wood combustion. The presence of bark, twigs and leaves have recently been shown to influence the emission of Ncontaining species during burning (Coggon et al., 2016), however, no difference in fuel composition is expected between the experiments as all bark and twigs were removed prior to combustion. As described above in a response to other Referee, differences in emission profiles may have been due to differences in ambient temperature effecting burner operation leading to differences in combustion rates.

L255 – Figure 2 is just mentioned here, and this could bear a bit more discussion and the differences between experiments discussed a bit further. One thing that stands out about these two experiments (2, 3) is that the CO2 loading in the chamber was substantially (almost half) lower than the other three experiments (despite the fact that the MCEs and many other quantities are essentially the same. If the injection time and dilution conditions were the same, this suggests that perhaps the combustion rate was lower (which would probably be indicated by lower flue temperature). Are there any other contextual or operational details that were different?

This point was raised by the other Referee as well. The MCE, which is very similar in all experiments, is unlikely to be the cause of the difference in emission profiles between the experiments. As great care was taken to replicate each burn as closely as possible (e.g., similar starting wood mass, number of logs/kindling pieces and wood arrangement prior to ignition) and experimental conditions (e.g., dilution factors), there is no obvious explanation for the interexperimental variability. The burner was housed in an uninsulated building and, as suggested by the both Referee, the variability could be due to effects of differences in outdoor temperature on the burner and chimney which would reduce the combustion rate. Each fire was allowed to burn for 15-20 minutes prior to injecting emissions into the smog chamber, which allowed the burner to warm up, however, we did not make temperature measurements in the burner or chimney and temperature differences may have remained. The discussion on inter-burn variability was expanded to include these points (Page 13, lines 262-266): "The burner is housed in an uninsulated building and the emission profile variability could be due to effects of outdoor temperature variability on the burner. For example, emission profiles from burning lignite and pyrolysis of bark and other biomass sources have been shown to vary with burn temperature (Hansson et al., 2004; Šyc et al., 2011)."

**L283 – This is too broad/definitive of a statement to make based on the narrow set of conditions tested here.**

The text has been modified as follows (Page 14, lines 304-306): "...making acetonitrile a poor marker for residential wood combustion under these burning conditions. Coggon et al. (2016) concluded that acetonitrile may not be a good tracer for residential burning in urban areas."

**L317-319 – It would be useful/instructive to attempt a mass balance on the NMOG and SOA loadings to estimate how much of the measured NMOG may be ending up in the condensed phase in your experiments.**

The work detailed in our previous publication on these experiments provides the first quantitative closure of the mass balance of the gas-phase species contributing to SOA (Bruns et al., 2016). We determined that the conversion of NMOGs traditionally included in models to SOA account for only ~3-27% of the observed SOA, whereas ~84-116% of the SOA can be explained by inclusion of non-traditional precursors, including naphthalene and phenol. The text was modified as follows (Page 16, lines 343-346): "Previous investigation of these experiments determined that the conversion of NMOGs traditionally included in models to SOA accounts for only ~3-27% of the observed SOA, whereas ~84-116% of the SOA accounts for only ~3-27% of the observed SOA, whereas ~84-116% of the SOA is explained by inclusion of non-traditional precursors, including naphthalene and phenol (Bruns et al., 2016)."

L358 – Were terpenes actually quantified? I don't expect much from birch wood, but if you measured them (or found them BDL) this should be noted.

As expected from previous studies (e.g., Schauer et al., 2001), monoterpenes were below the detection limit in all experiments. Isoprene emissions (Table 2) were also relatively low, although above the detection limit. This information has been added to the manuscript (Page 18, lines 394-395): "Monoterpene concentrations are below the detection limit in all experiments and isoprene emissions are relatively low (Table 2)."

L362 – 'Good agreement' is a bit vague, there is not-great agreement in panels c) and f). Could this indicate possibly misattribution of these compounds? For this figure, it would be helpful to show smoothed data (and probably a log y-axis) to make this a bit more readable.

We have modified the Figure (now Figure S3) to show the smoothed data (10 s data smoothed to 5 min moving average) to improve readability. We have also added some additional discussion of this Figure to address the Referee's comments that the agreement is better for some compounds (i.e., panels a, b, c, e, g, h) compared to others (i.e., panels d, f and i) (Page 19, lines 400-408): "There is generally good agreement between the observed and calculated decay for each compound which supports the structural assignment of each ion. For 2-methoxyphenol and 2,6-dimethoxyphenol (Figure S3 f and i, respectively), the agreement between the observed and calculated decays is not as good as for the other compounds, with slower decays than predicted. This discrepancy may be due to fragmentation of related compounds to form 2-methoxyphenol and 2,6-dimethoxyphenol in the instrument or formation of these compounds in the chamber during oxidation. For *o*-benzenediol, the decays are initially faster than expected and then become slower with increased aging, possibly due to the presence of isomers with different reaction rates with respect to OH."

**L365-370 - Somewhat confusing lead-in to discussion of Figure 3, as discussion emphasizes differences between experiments and this figure shows averages across all experiments - may make sense to just discuss this result then discuss inter-experiment differences (using Fig. 2)**

We agree with the Referee and have introduced Figure 3 in the preceding paragraph and then discussed the inter-experiment differences on SOA formation potential using Figure 2 in the next paragraph (Pages 18-20, lines 396-433): "We have previously identified the compounds contributing to the majority of the SOA formed during these experiments (Bruns et al., 2016). The average EF for each of these species is shown in Figure 3. Figure S3 shows the observed decay of the SOA precursors contributing the most to SOA formation during aging in the chamber compared to the expected decay based on the OH concentration in the chamber and the reaction rate with respect to OH. There is generally good agreement between the observed and calculated decay for each compound which supports the structural assignment of each ion. For 2-methoxyphenol and 2,6-dimethoxyphenol (Figure S3 f and i, respectively), the agreement between the observed and calculated decays is not as good as for the other compounds, with slower decays than predicted. This discrepancy may be due to fragmentation of related compounds to form 2-methoxyphenol and 2,6-dimethoxyphenol in the instrument or formation of these compounds in the chamber during oxidation. For *o*-benzenediol, the decays are initially faster than expected and then become slower with increased aging, possibly due to the presence of isomers with different reaction rates with respect to OH.

As described above, the overall primary emission profiles, as well as total NMOG emissions, vary considerably for experiments 2 and 3 compared to experiments 1, 4 and 5, with considerably higher total NMOG emissions in experiments 2 and 3. To determine the impact of the high NMOG emission experiments (2 and 3) compared to the lower NMOG emission experiments (1, 4 and 5) on SOA formation potential, individual SOA precursors with published SOA yields are investigated. The SOA formation potential for each of these 18 compounds is determined as the product of the primary EF and the best estimate SOA yield determined from the literature, as determined previously (Bruns et al., 2016). The total SOA formation potential for each experiment is taken as the sum of the individual SOA formation potentials. Interestingly, the SOA formation potential is similar in all experiments and the average enhancement of SOA formation potential in experiments 2 and 3 compared to the average of experiments 1, 4 and 5 is insignificant (Figure 2), despite the considerably different total NMOG EFs."

**References**

Bruns, E. A., El Haddad, I., Slowik, J. G., Kilic, D., Klein, F., Baltensperger, U., and Prévôt, A. S. H. (2016). "Identification of significant precursor gases of secondary organic aerosols from residential wood combustion." Scientific Reports, 6, 27881.